# CALIBRATED DECISION-MAKING THROUGH LARGE LANGUAGE MODEL-ASSISTED RETRIEVAL

## ABSTRACT

Recently, large language models (LLMs) have been increasingly used to support various decision-making tasks, assisting humans in making informed decisions. However, when LLMs confidently provide incorrect information, it can lead humans to make suboptimal decisions. To prevent LLMs from generating incorrect information on topics they are unsure of and to improve the accuracy of generated content, prior works have proposed Retrieval Augmented Generation (RAG), where external documents are referenced to generate responses. However, traditional RAG methods focus only on retrieving documents most relevant to the input query, without specifically aiming to ensure that the human user's decisions are well-calibrated. To address this limitation, we propose a novel retrieval method called Calibrated Retrieval-Augmented Generation (CalibRAG), which ensures that decisions informed by the retrieved documents are well-calibrated. Then we empirically validate that CalibRAG improves calibration performance as well as accuracy, compared to other baselines across various datasets.

## 1 INTRODUCTION

Large language models (LLMs; Jiang et al., 2023; Touvron et al., 2023; Dubey et al., 2024; Achiam et al., 2023) have demonstrated remarkable performance on numerous downstream natural language processing (NLP) tasks, leading to their widespread integration into various decision-making processes (Bommasani et al., 2021; Band et al., 2024; Zhou et al., 2024). However, even with significant increases in model size and the expansion of training datasets, it remains infeasible for LLMs to encode all possible knowledge within their parameters. As a result, the outputs produced by LLMs may not consistently be reliable for important human decision-making processes, potentially overlooking key or hidden details. Additionally, LLMs frequently provide inaccurate or misleading information with a high degree of confidence, a phenomenon referred to as *hallucination* (Zhuo et al., 2023; Papamarkou et al., 2024), which can lead humans to make flawed decisions. In addition, Zhou et al. (2024) have empirically demonstrated that human users often over-rely on LLM outputs during decision-making processes, and this over-reliance tends to increase in proportion to the model's confidence. Here, the model's confidence refers to the verbalized expression of how certain the model is when asked how confident it is in its answer. Specifically, they have found that for answers with high confidence, users show strong over-reliance regardless of whether the answer is correct or not. These findings highlight that utilizing LLMs without proper calibration of their responses and addressing the frequent occurrence of hallucinations can lead to incorrect decisions in high-stakes tasks like medical diagnosis and legal reasoning, potentially resulting in severe consequences (Li et al., 2019; 2022b; Han et al., 2024).

Retrieval Augmented Generation (RAG) (Lewis et al., 2020; Li et al., 2022a; Wang et al., 2024) has emerged as a promising method to address hallucinations, which is one of the two key issues when using LLMs in decision-making (Shuster et al., 2021; Li et al., 2024). Instead of generating answers directly, RAG retrieves relevant documents from external databases and uses them as an additional context for response generation. This approach supplements the information that LLMs lack, resulting in more accurate and reliable responses. However, the database cannot encompass all information, and the world knowledge is continuously being updated. In such cases, the retriever may retrieve irrelevant documents, which can distract the LLM and lead to the generation of incorrect answers to the question (Shi et al., 2023). Moreover, as described in Section 2.2, due to

the LLM's overconfidence in the retrieved document, they still tend to assign high confidence to its responses even when they are incorrect.

To address the issue of deep neural networks generating overconfident outputs for given inputs and to promote well-calibrated predictions, research on *uncertainty calibration* has been actively conducted across various fields (Kuleshov et al., 2018; Laves et al., 2020; Kapoor et al., 2024). In particular, for image classification tasks in computer vision, numerous techniques (Lakshminarayanan et al., 2017; Maddox et al., 2019; Thulasidasan et al., 2019) have been developed to improve uncertainty calibration. Especially, post hoc methods like temperature scaling, which simply adjust the output logits, have been shown to be simple yet effective in improving calibration (Kull et al., 2019; Vaicenavicius et al., 2019; Minderer et al., 2021; Widmann et al., 2022). However, in contrast to vision tasks, calibrating LLMs poses a more complex challenge due to their sequential token generation nature (Kapoor et al., 2024). Specifically, LLMs produce sequences of log probabilities for each token, and the number of possible sequences grows exponentially with length, making it impractical to apply traditional calibration methods that consider all output possibilities. This complexity renders straightforward adaptations of calibration techniques like temperature scaling ineffective for long-form sentence generation tasks in LLMs. To address these challenges, recent work by Band et al. (2024) proposed an uncertainty calibration method specifically designed for decision-making scenarios involving LLMs in long-form generation contexts. This method aims that the probabilities associated with user decisions, based on the guidance generated by the LLM, are well-calibrated. However, this method still lacks the ability to calibrate the probabilities associated with user decisions based on the guidance provided by RAG.

To address this issue, we propose the Calibrated Retrieval-Augmented Generation (CalibRAG) framework. CalibRAG allows an LLM using RAG to not only select relevant information to support user decision-making but also provide confidence levels associated with that information by utilizing a forecasting function, ensuring well-calibrated decisions based on the retrieved documents. Here, the forecasting function is the surrogate model that predicts the probability of whether the user's decision based on the guidance provided by RAG will be correct. We empirically validate that our CalibRAG significantly improves calibration performance as well as accuracy, compared to other relevant baselines across several datasets.

Our contributions can be summarized as follows:

- We propose the CalibRAG framework, which enables well-calibrated decision-making based on the guidance provided by RAG.
- We construct a new dataset by creating labels that indicate whether decisions made using retrieved documents correctly answer the questions, essential for training the forecasting function.
- We outperform existing uncertainty calibration baselines across various tasks involving RAG context using the Llama-3.1 model in decision-making scenarios.

## 2 PRELIMINARIES

### 2.1 DECISION CALIBRATION OF LONG FORM GENERATION

As discussed in Section 1, since human decision-makers tend to over-rely on the outputs of LLMs during the decision-making process, it is crucial to ensure that the confidence in LLMs' outputs is well-calibrated. To address this problem, Band et al. (2024) propose *decision calibration*, which aims to align the confidence of the model's predicted output with the accuracy of the user's decision based on the model output. This allows the user to make a reliable decision based on the model's confidence. Therefore, to achieve this goal, we need to ensure that the model not only generates factual information but also that its confidence in the generated responses accurately reflects the likelihood of correctness.

To formalize the problem, we introduce the following notations. Let $x \in \mathcal{X}$ represent the question or task for which a user needs to make a decision (*e.g.*, "What was the name of the 1996 loose adaptation of William Shakespeare's Romeo & Juliet written by James Gunn?"), and let $y \in \mathcal{Y}$ denote the corresponding true answer (*e.g.*, "Tromeo and Juliet"). Here, $\mathcal{X}$ and $\mathcal{Y}$ are the set of all possible questions and answers, respectively. Given the question $x$, the user provides an open-ended

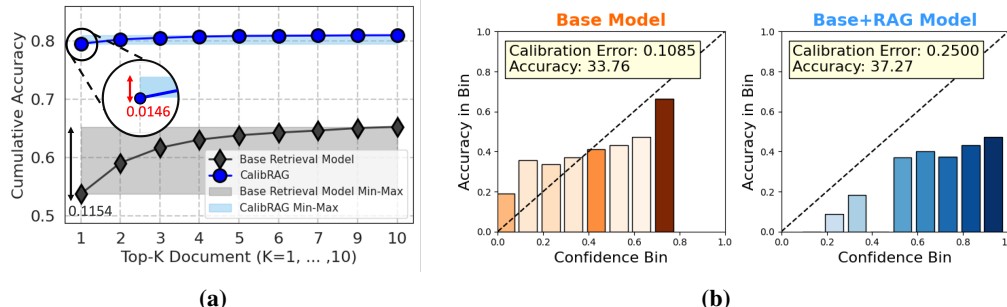

**Figure 1:** **(a) Cumulative accuracy using the top-10 documents shows an 11% improvement, demonstrating that top-1 document is not always optimal. CalibRAG achieves higher top-1 accuracy, with only marginal gains thereafter.** The base retrieval model is `contriever-msmarco`, evaluated on synthetic valid data (see Section 3.3).**(b)** Accuracy and calibration error on the NaturalQA dataset. **RAG outperforms the base model (orange) in accuracy (blue) but exhibits increased calibration error.** Bar height represents average accuracy per confidence bin, with darker shades indicating a higher density of predictions. The base model is `Llama-3.1-8B`, fine-tuned using the Number-LoRA method.

query $q(x)$ (*e.g.*, "Please provide an overview of the various adapted versions of Romeo and Juliet.") to an LLM as a prompt to gather information for the decision making about $x$. The LLM, denoted as $\mathcal{M}$, generates a long-form response to the query, *i.e.*, $z \sim \mathcal{M}(z|q(x))$, which serves as the guidance for the decision-making process. For the sake of notational simplicity, unless specified otherwise, we will use $q$ in place of $q(x)$. Given the question $x$ and the generated response $z$, the user leverages a forecasting function $f : \mathcal{X} \times \mathcal{Z} \to \Delta_{|\mathcal{Y}|}$ to assess all possible answers $y \in \mathcal{Y}$, where $\Delta_{|\mathcal{Y}|}$ denotes a simplex over the set $\mathcal{Y}$ and $\mathcal{Z}$ is the space of all possible responses from $\mathcal{M}$. The goal is to use the forecasting function $f$ to ensure that, given the long-form generated LLM response $z$, the user makes calibrated decisions on the question-answer pairs $(x, y)$. Based on this, Band et al. (2024) introduces formal definitions for three types of calibrations with varying conditions. For instance, the LLM is *confidence calibrated* (Guo et al., 2017) with respect to the forecasting function $f$ if $f$ is calibrated on the joint distribution $p(x, y, z)$, that is,

$$\Pr\Big(y = \arg\max_{j \in |\mathcal{Y}|} f(x, z)_j \mid \max_{j \in |\mathcal{Y}|} f(x, z)_j = \beta\Big) = \beta, \quad \forall \beta \in [0, 1], \tag{1}$$

where $f(x, z)_j$ denotes the $j^{\text{th}}$ element of the vector $f(x, z)$.

However, the method proposed by Band et al. (2024) to tackle this calibration problem has three major limitations: 1) it requires supervised fine-tuning for three different LLMs, including the LLM responsible for generating a response $z$ and the forecasting function $f$ parameterized with two LLMs, 2) it further needs proximal policy optimization (PPO; Schulman et al., 2017) for fine-tuning the LLM for response generation, which is known to suffer from training instability (Zhu et al., 2023), and 3) it cannot calibrate the probabilities associated with the user decisions based on the guidance provided by RAG.

## 2.2 RETRIEVAL AUGMENTED GENERATION (RAG)

Retrieval augmented generation (RAG) is first proposed by Lewis et al. (2020) and uses dense passage retrieval (DPR; Karpukhin et al., 2020) to retrieve and rank relevant paragraphs in question-answering (QA) tasks. The bi-encoder structure of the DPR model embeds questions and documents separately, enabling to precompute document embeddings and cache them in a vector database. A question is only embedded when presented and the similarity between the question and document embeddings is computed. The most relevant documents are retrieved and provided as additional context for the question to an LLM. The retrieved documents can guide the LLM to generate more reliable answers, rahther than solely relying on the knowledge encoded in its parameters.

Although RAG improves accuracy, retrieval models can still produce errors. First, since retrieval models are typically trained in an unsupervised manner (Izacard et al., 2021; Jin et al., 2023), the order of query-document similarities they produce does not necessarily align with how helpful those documents are for downstream user decisions. As shown in Figure 1a, the top-1 document retrieved

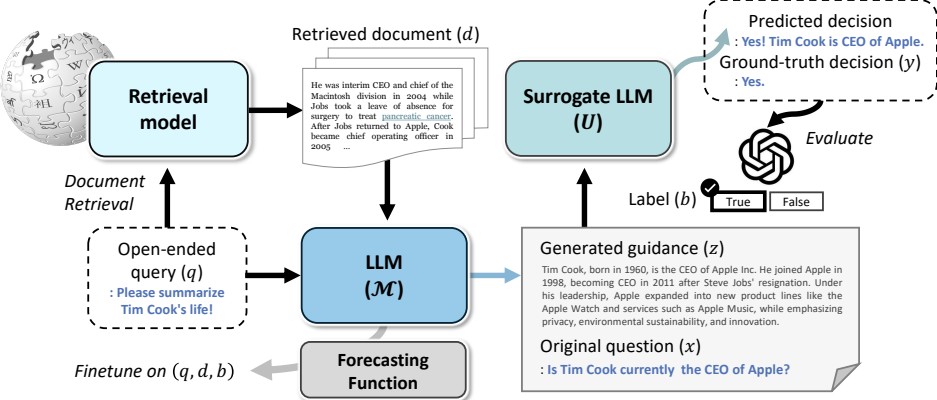

**Figure 2:** Overview of data generation and training process.

by the base retrieval model often leads to incorrect decisions. Rather subsequent documents could potentially improve the outcome. This indicates that the similarity scores assigned by the retrieval model do not always correlate with their utility in aiding decision-making. Additionally, RAG using an incorrect document may lead to flawed decision-making, as the LLM could introduce misleading information from irrelevant documents. As shown in Figure 1b, while RAG improves accuracy, the calibration error increases due to the tendency of the LLM to over-rely on irrelevant documents provided as context. Current RAG models do not address the confidence of the retrieved document.

To address these two issues, it is important to not only identify documents more relevant to downstream users through an additional reranking process but also to calibrate the confidence level of the retrieved documents.

## 3 CALIBRAG: RAG FOR DECISION CALIBRATION

**Overview.** We summarize our method and describe it in more detail in the following section. Given a task $x$ on which users make a decision and an open-ended query $q$ about the task, a retriever model gets a document $d$ relevant to the query from an external database. Based on the query and retrieved document, an LLM generates a guidance $z$ in the form of long-form generation that can help the user make an informed decision and outputs confidence $c$ for its response. To allow the LLM to express its uncertainty, we prompt the model to respond using either an integer number between 0 and 10 or linguistic terms of certainty (*e.g.*, "Ambiguous"). Finally, the user makes a final decision about the task, using both the guidance response $z$ and the LLM's confidence $c$. Our goal is to align the model confidence with accuracy of the user's decision based on the guidance. To this end, we train a forecasting function $f(q, d)$ that gets the query and retrieved document as input predicts the probability of the decision being correct, and uses it as a ranking function of the retriever model. The overall pipeline of our method is illustrated in Figure 2.

### 3.1 PROBLEM SETUP

Following Band et al. (2024), to train and evaluate the forecasting function, we first use an LLM surrogate model, denoted as $U$, to mimic human decision-making when making decisions, instead of relying on actual human users. However, unlike Band et al. (2024), we leverage the human evaluation results from Zhou et al. (2024) to design a prompt that steers the surrogate LLM model $U$ to exhibit more human-like behavior. Specifically, the prompt is crafted to lead the surrogate $U$ to place strong belief in the confidence of the LLM, denoted as $\mathcal{M}$, generating the guidance in its responses. For further details on the prompt, please refer to Appendix F. Additionally, since the user's decision (mimicked by $U$) is usually given as a free-form text rather than being a simple class label, we use GPT-4o-mini (Achiam et al., 2023) model, denoted as $\mathcal{G}$, to evaluate the correctness of the user's decision compared to the true answer $y$.

Let $x$ be a task and $q$ the corresponding open-ended query. The retriever model retrieves a document $d$ from the external database $\mathcal{E}$. Then, the LLM model $\mathcal{M}$, responsible for generating the guidance based on $d$, takes both the query $q$ and the retrieved document $d$, and produces the guidance and confidence $[z, c] \sim \mathcal{M}(z, c|q, d)$ for the decision-making task $x$. As we discussed in Section 1 and illustrated in Figure 1b, the RAG model $\mathcal{M}$ often generates a guidance $z$ with overly-high confidence $c$. This can lead to miscalibrated predictions when the confidence does not accurately reflect the correctness of the generated information $z$. Following this, our user model $U$ makes a decision $U(x, z, c)$ by utilizing the question $x$, the guidance $z$, and the confidence $c$. Then our final goal is to learn a forecasting function $f : \mathcal{Q} \times \mathcal{E} \to [0, 1]$ defined on a product space of query space $\mathcal{Q}$ and the external dataset $\mathcal{E}$ and satisfies the following *binary* calibration equation,

$$\mathbb{E}\left[\mathcal{G}(y, U(x, z, f(q, d)))|f(q, d) = \beta\right] = \beta, \quad \forall \beta \in [0, 1]. \tag{2}$$

where $\mathcal{G}(y, U(x, z, f(q, d))) \in \{0, 1\}$ is a binary variable indicating whether a user decision $U(x, z, f(q, d))$ matches the answer $y$. Note here that we are using the forecasted confidence $f(q, d)$ in place of the confidence $c$ generated from $\mathcal{M}$. This adjustment means that forecasting function $f$ is designed to predict the probability of a correct answer for a given task $x$, the guidance $z$, and the confidence derived from $f$, by utilizing the query $q$ and the retrieved document $d$. We expect that the actual accuracy will be well-aligned with the predicted probabilities, ensuring a well-calibrated decision-making process.

## 3.2 Modeling and Training

To model the forecasting function $f$, it is essential to have the capacity to sufficiently analyze the relationship between the query $q$ and the retrieved document $d$. For this reason, we use a pre-trained LLM encoder $f_{\text{feat}}$ as the base feature extractor model. Additionally, to model the probability of whether $U(x, z, f(q, d))$ is correct or not, we attach a linear classifier head after $f_{\text{feat}}$. This head uses a sigmoid function on the logits to generate the probability values. For efficient learning during supervised fine-tuning, we keep the weights of the pre-trained $f_{\text{feat}}$ fixed and employ Low-Rank Adaptation (Lora; Hu et al., 2021) to train the feature extractor. This allows us to adapt the model efficiently with minimal additional parameters. Then our overall forecasting function $f$ is formulated as follows:

$$\Pr(\mathcal{G}(y, U_f) = 1) = f(q, d) \coloneqq \text{sigmoid}\left(W_{\text{head}}^{\top} f_{\text{feat}}(\text{concat}[q, d]; W_{\text{LoRA}}) + b_{\text{head}}\right) \tag{3}$$

where sigmoid and concat denote the sigmoid function $x \mapsto 1/(1 + \exp(-x))$ and the concatenate operation, respectively. $W_{\text{head}}$, $b_{\text{head}}$, and the LoRA weight $W_{\text{LoRA}}$ are learnable parameters, and $U_f$ is the shorthand for $U(x, z, f(q, d))$. Here, the reason $f$ can model $p(\mathcal{G}(y, U_f) = 1)$ using only the query $q$ and document $d$ is that $q$ and $d$ depend on $x$, and $z$ also depends on both $q$ and $d$. This enables $f$ to acquire enough information from the query and the retrieved document to forecast the distribution of correctness of the decision $y$. To train our forecasting function $f$, we employ a synthetic dataset whose construction will be described in the next section. The model is trained with the following binary cross-entropy loss,

$$\mathcal{L} = -\frac{1}{|\mathcal{T}|} \sum_{(q,d,b) \in \mathcal{T}} (b \log f(q, d) + (1 - b) \log(1 - f(q, d))) \tag{4}$$

where $\mathcal{T}$ represents the synthetic training dataset, and $b \in \{0, 1\}$ is a binary label indicating the correctness of the user's decision. Here, through supervised learning using various combinations of $q$, $d$, and $b$, the trained function $f$ can analyze the relationship between unseen combinations of $q$ and $d$ using the learned feature map, enabling it to predict the probability of the decision.

## 3.3 Synthetic Supervision Data Generation

To conduct the supervised learning discussed in Section 3.2, it is essential to construct an appropriate synthetic training dataset $\mathcal{T}$ consisting of the triples $(q, d, b)$. We first extract the $(x, y)$ (*e.g.*, ("In which county is Ascot", "Berkshine, England")) decision-making task pairs from the following three Question Answering datasets: 1) TriviaQA (Joshi et al., 2017), 2) SQuAD2.0 (Rajpurkar et al., 2018), and 3) WikiQA (Yang et al., 2015) datasets. Then, for every $x$ in the training dataset, we generate an open-ended query $q$ (*e.g.*, "Write a paragraph about the county where Ascot is located.") based on each $x$, using the GPT-4o-mini model. At this point, it is important to note that instead

of retrieving only the single top document $d$ with the highest similarity score from the retriever model for each query $q$, we retrieve the top 20 documents. There are two reasons for this. First, as illustrated in Figure 1a, a large number of low-ranked documents actually help the surrogate user make a correct decision. If we only include the top-1 documents, many of which would be labeled as incorrect, the synthetic dataset would be highly biased to negative samples. Second, using only one $d$ per $(x, y)$ pair for labeling and training could result in the model overfitting to the label without learning the relationship between $q$ and $d$ adequately. By pairing the same $q$ with various $d$'s, the model can learn from positive and negative samples, improving its ability to generalize. After retrieving multiple documents, we provide each $(q, d)$ pair to the RAG model $\mathcal{M}$, which generates the guidance $z$ based on $d$ (*e.g.*, "Ascot, Berkshire") and a certainty level $c$ (*e.g.*,"Certainty: 9"). Then, the triple $(x, z, c)$ is passed to the user model $U$. The model's decision is compared with the true answer $y$ by the evaluation function $\mathcal{G}$, which determines whether the decision is correct, and this is recorded as a binary label $b$. Thus, for each $(x, y)$ pair, we can generate 20 different training triples $(q, d, b)$. Refer to Appendix D for examples of 20 different retrieved documents and their corresponding labels.

## 3.4 Inference

After finishing the training of the forecasting function $f$, we perform inference for a new decision task $x^*$ through the following four stage process:

**Stage 1: Initial retrieval of documents**. Given an open-ended query $q^*$, derived from the original question $x^*$, we begin the document retrieval process using the retrieval model. Similar to the training data generation process, we retrieve the top $K$ relevant documents from the external database, denoted as $\mathcal{D}^* := \{d_i^*\}_{i=1}^K$. The goal of this stage is to construct a diverse set of candidate documents that may contain valuable information for producing the correct answer $y$.

**Stage 2: Scoring and selection of documents.** Once the $K$ candidate documents are retrieved, we predict the decision confidence level for each document using our trained forecasting function $f$. At this point, regardless of the similarity score from the retrieval model, each document is assigned a new rank based on its confidence level predicted with $f$. Specifically, the ranking is determined based on the probability that the user will make a correct decision when provided with the guidance generated from each document, with documents arranged in descending order of the forecasted probabilities $\{f(q^*, d_i^*)\}_{i=1}^K$. The document with the highest ranking is selected for the next stage. Here, if the predicted probability for the highest-ranked document $d^*$ is lower than a pre-defined threshold $\epsilon$ (with more details about $\epsilon$ provided in Appendix B), we set this probability to 0.5. In such cases, we determine that none of the currently retrieved $K$ documents are useful for assisting with the decision task $x^*$. Consequently, in this case, we proceed to Stage 3 to retrieve a new set of $K$ candidate documents. If this condition is not met, we move forward to Stage 4.

**Stage 3: Reformulating the query.** If the predicted probability for the highest-ranked document $d^*$ is lower than a pre-defined threshold $\epsilon$ in Stage 2, to retrieve a new set of $K$ candidate documents, we reformulate our open-ended query $q^*$ into $q^{**}$ by emphasizing more important content from the question $x$. This reformulation focuses on extracting key aspects of the original task, ensuring that the next retrieval attempt targets more relevant and helpful documents. After reformulating the query, we repeat Stage 1 and Stage 2 once again. Examples of query reformulation are shown in Appendix C.

**Stage 4: Final decision.** After retrieving the document $d^*$, we generate the guidance $z^*$ using the RAG model $\mathcal{M}$. The user model $U$ then makes a decision $U(x^*, z^*, f(q^*, d^*))$. This decision is compared with the correct answer $y^*$ by $\mathcal{G}$ to determine its accuracy.

## 4 Experiments

### 4.1 Setup

**Implementation detail.** For all experiments, following Section 3.3, we collect a total of 20,870 samples for training and 4,125 for validation. We employ the `Llama-3.1-8B` (Dubey et al., 2024) model as both the RAG model $\mathcal{M}$ and decision model $U$. For evaluating the long-form generated answers, we utilize the `GPT-4o-mini` API as an evaluation model $\mathcal{G}$. Additionally, we

**Table 1:** Comparison of zero-shot evaluation of calibration baselines across multiple datasets. † indicates one additional regeneration step if the confidence does not reach the threshold. **CalibRAG demonstrates a lower no-answer rate while achieving higher accuracy and lower calibration error compared to other baselines.**

| Methods/Dataset | BioASQ | | | | | | HotpotQA | | | | | |
|---|---|---|---|---|---|---|---|---|---|---|---|---|
| | AUROC (↑) | ACC (↑) | ECE (↓) | BS (↓) | NLL (↓) | %NA | AUROC (↑) | ACC (↑) | ECE (↓) | BS (↓) | NLL (↓) | %NA |
| *Base* | - | 27.41 | - | - | - | 26.10 | - | 28.47 | - | - | - | 41.82 |
| *CT-probe* | 58.11 | 28.19 | 0.3368 | 0.3559 | 1.1195 | 28.05 | 55.95 | 31.75 | 0.3600 | 0.3773 | 1.2479 | 32.42 |
| *CT-LoRA* | 65.74 | 29.05 | 0.3664 | 0.3640 | 1.1729 | 26.83 | 60.87 | 30.13 | 0.3858 | 0.3842 | 1.4122 | 37.29 |
| *Number-LoRA* | 65.40 | 28.84 | 0.2677 | 0.2992 | 0.8220 | 32.43 | 63.91 | 26.54 | **0.1971** | **0.2643** | **0.7724** | 50.53 |
| *Linguistic-LoRA* | 51.72 | 31.02 | 0.2868 | 0.3828 | 1.0311 | **24.28** | 51.07 | 33.64 | 0.2886 | 0.3100 | 0.9413 | 33.76 |
| *CalibRAG* | 71.21 | 35.03 | **0.2500** | **0.2900** | **0.7899** | 27.31 | 65.47 | 39.28 | 0.2414 | 0.2876 | 0.8276 | 26.85 |
| *CalibRAG†* | **76.50** | **35.98** | 0.2667 | 0.2779 | 0.7560 | 25.16 | **68.51** | **40.70** | 0.2392 | 0.2642 | 0.7390 | **25.90** |

| Methods/Dataset | WebQA | | | | | | NQ | | | | | |
|---|---|---|---|---|---|---|---|---|---|---|---|---|
| | AUROC (↑) | ACC (↑) | ECE (↓) | BS (↓) | NLL (↓) | %NA | AUROC (↑) | ACC (↑) | ECE (↓) | BS (↓) | NLL (↓) | %NA |
| *Base* | - | 35.84 | - | - | - | 31.16 | - | 36.95 | - | - | - | 30.90 |
| *CT-probe* | 57.94 | 37.31 | 0.3572 | 0.3724 | 1.2797 | 13.03 | 58.15 | 38.53 | 0.3898 | 0.4273 | 1.3313 | 14.93 |
| *CT-LoRA* | 62.54 | 33.48 | 0.3382 | 0.3507 | 1.0852 | 14.18 | 64.08 | 38.89 | 0.3936 | 0.3670 | 1.2574 | 18.05 |
| *Number-LoRA* | 63.55 | 35.05 | 0.3382 | 0.3372 | 0.9688 | 15.32 | 64.83 | 38.40 | 0.2608 | 0.2914 | 0.8707 | 24.23 |
| *Linguistic-LoRA* | 50.33 | 36.88 | 0.4894 | 0.4809 | 1.3977 | 9.45 | 51.37 | 41.44 | 0.4220 | 0.4254 | 1.2433 | 11.86 |
| *CalibRAG* | 69.58 | 44.32 | **0.3064** | 0.3251 | **0.8919** | 6.65 | 66.36 | 48.29 | **0.2596** | 0.2994 | 0.8490 | **8.39** |
| *CalibRAG†* | **73.23** | **45.03** | 0.3194 | **0.3113** | 0.9035 | **5.43** | **69.40** | **49.10** | 0.2625 | **0.2876** | **0.8150** | **8.39** |

used `Contriever-msmarco` (Izacard et al., 2021) as the base retrieval model. In all tables, the best performance is indicated with **boldfaced underline**, while the second-best value is represented with underline in each column.

**Baselines.** We compare CalibRAG with the following relevant baselines.

- **Uncertainty calibration baselines:** (1) *Calibration Tuning* (Kapoor et al., 2024) labels the correctness of the prediction $\hat{y}_i$ to the question $x_i$ and utilizes these triples $\{(x_i, \hat{y}_i, b_i)\}$ for fine-tuning. The two variants are **CT-probe**, which adds a classifier head to estimate the probability of the correctness of the prediction, and **CT-LoRA**, which outputs "Yes" or "No" to the question "Is the proposed answer true?" (2) *Verbalized Confidence Fine-tuning*, as used by (Tian et al., 2023; Xiong et al., 2024), samples multiple predictions $\hat{y}_{ik}$ for each $x_i$ and maps ratio of the correct answers into confidence level: either integer between 0 and 10 (**Number-LoRA**) or linguistic terms indicating uncertainty (**Linguistic-Lora**). At inference time, all the models use the top-1 document retrieved by the base retriever as additional context. Further details are in Appendix F.

- **Reranking baselines:** Although our method is primarily designed to verify the utility of context retrieved by the retrieval model and calibrate confidence, it can also be viewed as a reranking approach for retrieved documents in downstream tasks. Accordingly, we compare our model against various reranking methods: (1) **Base**, which uses the top-1 document without any reranking. (2) **Cross-encoder**, which reranks documents using a cross-sentence encoder that jointly embeds the query and document, and then outputs their similarity score. (3) **LLM-rerank** (Sun et al., 2023), which involves prompting the LLM to rerank by leveraging the relationship between the query $q$ and the documents $d$.

**Evaluation metrics.** We evaluate all the models in terms of accuracy, AUROC, and various calibration metrics such as Expected Calibration Error (ECE; Naeini et al., 2015), Brier Score (BS; Brier, 1950), and Negative Log Likelihood (NLL). Moreover, we measure the percentage of LLM abstaining from predicting an answer, denoted as %NA. Details regarding these metrics can be found in Appendix A.

**Zero-shot evaluation.** We also utilize BioASQ (Krithara et al., 2023), HotpotQA (Yang et al., 2018), WebQA (Chang et al., 2022), and NQ (Kwiatkowski et al., 2019) for zero-shot evaluation. In our comparison of the uncertainty calibration baselines, all uncertainty baselines employ the top-1 document $d_1^*$ from the **Base** retrieval model for the LLM $\mathcal{M}$ to generate the guidance $z^*$ related to the open-ended query $q^*$. In contrast, CalibRAG re-ranks the original top-20 documents with the forecasting function $f$ and selects the document with the highest confidence score of $f$ to produce the

**Table 2:** Comparison of reranking methods based on accuracy across different datasets under original RAG settings and direct RAG setting. In this setting, confidence was not incorporated into the decision process. CalibRAG consistently outperforms other reranking methods in terms of accuracy.

| Methods | BioASQ | HotpotQA | WebQA | NQ | Avg. |
|---|---|---|---|---|---|
| Base | 32.02 | 35.74 | 38.13 | 43.03 | 37.23 |
| Cross-encoder | 31.48 | 37.70 | 42.25 | 44.10 | 38.88 |
| LLM-rerank | 36.34 | 35.15 | 37.51 | 41.28 | 37.57 |
| CalibRAG | 37.57 | **43.84** | 45.03 | **49.85** | 44.07 |
| CalibRAG† | **37.61** | **44.16** | **45.97** | **49.90** | **44.41** |

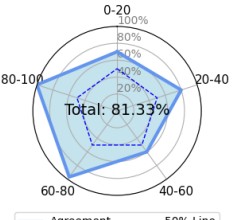

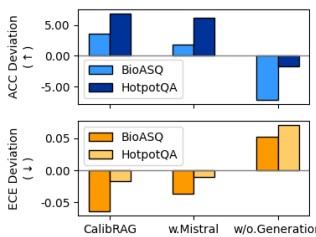

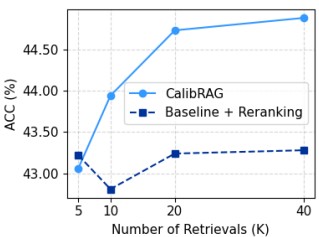

**(a)** Human-Model Agreement Rate  **(b)** Ablation of $z$ Generation  **(c)** Different number of retreival

**Figure 3:** (a) Agreement rates between human annotators and the model. (b) Performance impact when model generation is omitted. The 0.0 line represents the best baseline from Table 1. (c) The effect of varying the number of retrieved documents on reranking performance on WebQA dataset.

guidance $z^*$, as outlined in **Stage 2** of the inference process. The uncertainty baselines do not take into account the confidence of $d_1^*$; hence, we leverage the confidence $c^*$ generated by $\mathcal{M}$ concerning $q^*$ and $z^*$ for answer prediction. For CalibRAG, we generate the confidence using $f(q^*, d^*)$, and the surrogate user $U$ makes a decision based on $x^*$, $z^*$, and $f(q^*, d^*)$.

## 4.2 MAIN RESULTS

**Comparison with uncertainty calibration baselines.** Table 1 presents a comparison of uncertainty-based baselines across four QA datasets. Our CalibRAG achieves both a lower 'No Answer' rate and higher accuracy compared to other baselines, achieving the accuracy of 35.03 and 39.91 on BioASQ and HotpotQA, respectively, representing over a 3% improvement over the best-performing baseline. Additionally, its confidence level is better calibrated than the other baselines, demonstrating the lowest ECE and BS. CalibRAG†, which regenerates the query for documents that do not exceed the threshold, consistently shows performance improvements. However, while it correctly answers more challenging questions, it also makes accurate decisions with lower confidence, causing some variation in the calibration metrics.

**Comparison with reranking baselines.** For a fair comparison with the reranking baselines, we assume a scenario where the surrogate user $U$ makes decisions using only the question $x^*$ and the guidance $z^*$ without leveraging the confidence prediction $c^*$, *i.e.*, $U(x^*, z^*)$. In the case of CalibRAG, although the confidence predicted by the forecasting function $f$ is not provided to the user, the reranking is based on $f$'s prediction. This means that, unlike the other baselines, CalibRAG takes the confidence of $f$ into account for reranking. Table 2 highlights the reranking capability of CalibRAG, achieving an average accuracy improvement of 5.19% over the reranking with cross-encoder. Notably, CalibRAG† once again results in further performance improvement, similar to the previous experiment. In contrast, the LLM-rerank method even underperforms HotpotQA and NQ compared to the cross-encoder baseline due to cases where the LLM either refuses to answer or generates incorrect tokens. These findings demonstrate the superior performance of CalibRAG in reranking for RAG.

## 4.3 ABLATION STUDIES

In this section, we provide ablation studies to demonstrate the performance of CalibRAG.

**Does an LLM approximate human decision making?** Since it is impractical to directly hire human annotators to generate and evaluate large amounts of data, as mentioned earlier, we follow the setup of Zhou et al. (2024) by crafting prompts to encourage the LLM to mimic human decision behaviors. As illustrated in Fig. 3a, we ask each of 10 human annotators to answer 10 questions based on the guidance $z$ as well as the confidence level $c$ corresponding to specific confidence bins. The agreement rate exceeded $50\%$ in all confidence bins, achieving an average agreement rate of $81.33\%$. This indicates that our LLM serves as an effective surrogate through prompting, which is consistent with the results reported by Zhou et al. (2024).

**Does CalibRAG generalize to utilize unseen RAG models?** The way CalibRAG constructs the synthetic dataset $\mathcal{T}$, used for training the forecasting function, depends on the RAG model $\mathcal{M}$, which is responsible for generating the guidance $z$. In this experiment, we study how well our CalibRAG can generalize to utilize an unseen LLM as the RAG model for decision-making task. We use `Mistral-7B` for the RAG model and plot its performance improvement over the best-performing baseline. As shown in Fig. 3b, our CalibRAG with `Mistral-7B` still improves the accuracy and ECE, indicating the effectiveness of CalibRAG with the unseen RAG model. Compared to `Llama-3.1-8B`, it slightly underperforms due to inherent performance disparities between the two models.

**What is the effect of directly using retrieved documents for prediction?** In this experiment, we study the effectiveness of utilizing the guidance generated by the RAG model $\mathcal{M}$. To this end, instead of generating the guidance $z^*$ with respect to the query $q^*$, we directly provide the retrieved document $d^*$ to the surrogate user $U$ for prediction of the task $x^*$, *i.e.*, $U(x^*, d^*, f(q^*, d^*))$ instead of $U(x^*, z^*, f(q^*, d^*))$, and evaluate its performance. As illustrated in Fig. 3b, prediction without generating the guidance $z^*$, denoted as "w/o Generation", significantly degrades both accuracy and ECE. This degradation is attributed to irrelevant parts of the retrieved document that distract the surrogate user $U$, leading to an incorrect decision (Shi et al., 2023).

**How does the number of retrieved passages ($K$) impact reranking?** We use $K = 20$ documents for reranking in all the experiments, considering the trade-off between its computational cost and the performance of the decision-making task. To validate our choice, we plot accuracy as a function of the number of documents for reranking in Fig. 3c. The results show that performance improves up to 20 documents, but the gains diminished beyond 40 documents, supporting our choice of 20 documents. This indicates that the retrieval model gets most of the relevant documents in the initial stage, and a more advanced reranking would be necessary for further improvement.

## 4.4 Qualitative Results

While quantitative metrics alone may not fully capture all the benefits of CalibRAG, we present examples highlighting its ability to identify relevant documents and assign calibrated confidence scores. Given the query "Write a paragraph about the kind of bug that uses the American Sweetgum as a host plant.", the base retriever focuses only on the keyword "American Sweetgum,", retrieving loosely relevant content and marking its confidence as 'Confident' (10/11) as illustrated in Fig. 4. This led to the incorrect conclusion that the sweetgum is the host plant of Parcoblatta divisa, the southern wood cockroach. In contrast, CalibRAG captures the full context, retrieving documents specifically about the gypsy moth, which uses the sweetgum as a host plant, and correctly assigns a confidence level of 81.41. This demonstrates the capability of CalibRAG to find a relevant document and assign a confidence level correlated with the accuracy of the downstream surrogate user. Additional examples can be found in Appendix D.

## 5 Related Works

### 5.1 Uncertainty Calibration in Language Models

Traditional calibration techniques primarily rely on token-level log probabilities (Guo et al., 2017). However, many modern LLMs are autoregressive, allowing the generation of token sequences through the chain rule of probability by multiplying the conditional probabilities of each token (Achiam et al., 2023). To estimate the concept-level probability within such generated sen-

> **Original question:** The American Sweetgum is the hostplant of what kind of bug?
> **Open-ended query:** Write a paragraph about the kind of bug that uses the American Sweetgum as a host plant.
>
> **Answer:** moth

> **Original Retrieval Model's Top-1 document** (This context set the LM model's confidence to 'Confident'):
> … American sweet gum, and other deciduous trees), and trapped in molasses-baited jars. One researcher who collected specimens extensively found it to be the most adaptable of the "Parcoblatta" species, trapping adults among logs and undergrowth on the borders of woodland areas, and taking specimens from pasture grasses, in grass under backyard trees, under dried cow dung, under trash and debris at woodland campsights, and from homes in wooded areas, which the species is sometimes reported to invade. Parcoblatta divisa Parcoblatta divisa, the southern wood cockroach, is a species of cockroach native to the United States.
> **User decision: The American Sweetgum is the host plant of the Parcoblatta divisa, also known as the southern wood cockroach.**

> **CalibRAG's Top-1 document** (This context set the CalibRAG's confidence to 81.41):
> … sometimes as imitation mahogany or Circassian walnut. It is used widely today in flake and strand boards. Sweetgum is a foodplant for various Lepidoptera caterpillars, such as the gypsy moth. The American sweetgum is widely planted as an ornamental, within its natural range and elsewhere. The hardened sap, or gum resin, excreted from the wounds of the sweetgum, for example, the American sweetgum ("Liquidambar styraciflua"), can be chewed on like chewing gum and has been long used for this purpose in the Southern United States. The sap was also believed to be a cure for sciatica, weakness of nerves, etc.
> **User decision: The American Sweetgum is the host plant of the gypsy moth.**

**Figure 4:** Qualitative comparison of original retrieval model from CalibRAG.

tences, summing over all possible corresponding probabilities would be required—an intractable process due to the exponential number of potential sequences. Consequently, token-level probabilities in current language models often fail to offer reliable confidence estimates for long-form text generation, thereby limiting their application to tasks that extend beyond multiple-choice scenarios.

Recently, various prompting-based approaches have been explored to address this limitation, leveraging verbalized expressions to quantify uncertainty (Tian et al., 2023; Xiong et al., 2023). For instance, a model can be prompted with: *"Please indicate your confidence level in your answer by providing a number between 0 and 100."* If the model generates a response such as *"90"*, this value can be interpreted as the confidence level of its answer. However, when using zero-shot probabilities for uncertainty estimation, recent LLMs often display overconfidence in their predictions, leading to poorly calibrated outputs (Papamarkou et al., 2024). This remains a significant challenge in enhancing the reliability and robustness of LLMs for more complex decision-making tasks.

## 5.2 RERANKING FOR RETRIEVAL AUGMENTED GENERATION

RAG leverages external knowledge to produce accurate answers in Open-Domain QA. However, not all documents retrieved by the retrieval model hold the same importance, and many contain noise, making reranking essential to select the most relevant documents (Glass et al., 2022). LLM-based reranking is an effective approach as it captures complex semantic relationships between documents and queries to reorder the retrieved documents appropriately (Sun et al., 2023). Another prominent reranking method uses cross-encoders, which take both the question and document as input, considering their interactions to perform more precise reranking (Li et al., 2022c). These diverse reranking approaches help RAG systems minimize noise from retrievers and select the most pertinent information to generate optimal answers.

## 6 CONCLUSION

In this paper, we introduced CalibRAG, a simple yet effective framework designed to improve confidence calibration and ensure more reliable document retrieval. Our experiments demonstrated that CalibRAG significantly enhances QA performance within the RAG setting across various benchmark datasets. Moreover, ablation studies showed that CalibRAG effectively aligns model confidence with factual correctness, resulting in improved decision-making accuracy and calibration. Overall, CalibRAG stood out as a robust solution for enhancing the reliability of RAG-based LLM guidance in decision-driven scenarios. However, creating synthetic datasets and training the forecasting function for decision calibration may introduce some overhead. Nonetheless, accurately calibrating language model confidence is crucial, making this approach both valid and worthwhile.

**Reproducibility statement.** We present the overall dataset generation and training procedure in Fig. 2. Additionally, we further present all the details regarding experimental environments, datasets, hyperparameters, and evaluation metrics in Appendix A.

**Ethics statement.** In this paper, we proposed a method that enables well-calibrated decision-making based on the guidance provided by RAG. During the synthetic data generation process, we did not create or use datasets containing personal or sensitive information; instead, we processed existing publicly accessible document datasets to create new datasets, thus avoiding ethical issues. On the other hand, as various human users increasingly utilize LLMs in different aspects of daily life, the trustworthiness of LLM outputs is becoming increasingly important. We specifically enhanced the model's guidance by providing additional confidence in situations where users rely on LLMs for decision-making. This approach helps users trust the accuracy of the guidance, thereby offering a positive societal impact by increasing users' confidence in LLMs.

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

## A  Experimental details

Our implementation builds on key libraries such as PyTorch 2.1.2 (Paszke et al., 2019), Huggingface Transformers 4.45.1 (Wolf et al., 2019), and PEFT 0.7.1 [1], providing a robust foundation for experimentation. We employ the `Llama-3.1-8B-Instruct` model, a state-of-the-art open-source multilingual LLM available from Hugging Face models. [2] Our experiments are executed on high-performance NVIDIA RTX 3090 and RTX A6000 GPUs, ensuring efficient and scalable model training. Additionally, we utilize the official `facebookresearch-contriever` repository for our retrieval model[3]. For training baselines, we reference the calibration-tuning repository. [4]

### A.1  Datatsets

**Train Datasets**  SQuAD2.0 (Rajpurkar et al., 2018) is a reading comprehension dataset sourced from Wikipedia, containing questions answered by text spans from the articles, including some unanswerable ones. WikiQA (Yang et al., 2015) is a question-sentence pair dataset from Wikipedia, designed for open-domain question answering and includes unanswerable questions for research on answer triggering. TriviaQA (Joshi et al., 2017) is a reading comprehension dataset with questions authored by trivia enthusiasts, paired with evidence documents from Wikipedia and other web sources. We randomly sampled 10,000 data points each from TriviaQA and SQuAD, and collected all 873 training samples from WikiQA, resulting in a total of 20,873 training data. For the validation set, we gathered 2,000 samples each from TriviaQA and SQuAD, along with 126 samples from WikiQA, resulting in 4,126 validation data points. After removing null values, we compiled 20,870 training and 4,125 validation data. For CalibRAG, we retrieved the top 20 documents for each query from a set of 21,015,300 Wikipedia articles. we downloaded all these datasets in Hugging Face datasets. [5] For construction of labeled dataset $\mathcal{T}$ used to train the forecasting function of CalibRAG, we collect positive and negative documents for each query as follows. If the first correct document is ranked at position $k$, the top $k - 1$ documents are labeled as negative and the correct document is labeled positive. Each $k$ documents are paired with corresponding query and added to the dataset $\mathcal{T}$. If we find the correct document ranked at position 1, only the correct document is added to the dataset. This process resulted in a total of 27,220 training data points and 6,271 validation data points.

**Evaluation Datasets**  For zero-shot evaluation, we employ several datasets covering diverse domains and question types. BioASQ (Krithara et al., 2023) is a biomedical QA dataset containing factoids, lists, and yes/no questions derived from PubMed articles. HotpotQA (Yang et al., 2018) is a multi-hop question-answering dataset requiring reasoning across multiple supporting documents from Wikipedia to find answers, emphasizing a more complex retrieval and reasoning process. We-bQA (Chang et al., 2022) is an open-domain question-answering dataset consisting of natural, conversational questions paired with web documents, targeting real-world, context-rich scenarios. Natural Questions (NQ) (Kwiatkowski et al., 2019) is another large-scale question-answering dataset, designed to answer questions based on Wikipedia articles, containing both long-form and short-form answers. These datasets are used without additional training, providing a robust evaluation of the generalization capabilities of CalibRAG across different domains and question types.

### A.2  Hyperparameters

Table 3 outlines the hyperparameters used for training the base model and LoRA, including key parameters such as learning rate, batch size, and LoRA-specific settings like rank and alpha.

---

[1] https://github.com/huggingface/peft
[2] https://huggingface.co/meta-llama/Meta-Llama-3.1-8B-Instruct
[3] https://github.com/facebookresearch/contriever
[4] https://github.com/activatedgeek/calibration-tuning
[5] https://github.com/huggingface/datasets

**Table 3:** Hyperparameters for LLM Training

| Base Model Hyperparameters | | LoRA Hyperparameters | |
|---|---|---|---|
| **Hyperparameter** | **Value** | **Hyperparameter** | **Value** |
| Learning Rate | $[10^{-4}, 10^{-5}]$ | LoRA Rank | 8 |
| Batch Size | $[1, 4]$ | LoRA Alpha | 16 |
| Max Steps | 10,000 | LoRA Dropout | 0.1 |
| Optimizer | AdamW | | |
| Dropout Rate | 0.0 | | |
| Gradient Accumulation Steps | $[1, 4]$ | | |
| Weight Decay | 0.01 | | |
| Gradient Clipping | 1.0 | | |
| Warmup Steps | 500 | | |
| Scheduler | Linear | | |

A combination of which two drugs was tested in the IMbrave150 trial?  *

Generated Context: The combination of drugs tested in the IMbrave150 trial was atezolizumab (an anti-PD-L1 antibody) and bevacizumab (an anti-VEGF antibody).

Model Confidence: **44.49 (0-100)**

○  Combination of atezolizumab and bevacizumab

○  No answer

**Figure 5:** Human evaluation format

## A.3 EVALUATION METRICS

To evaluate long-form text, we utilized `gpt-4o-mini` to compare the ground-truth answers with the predicted answers in all cases. Based on this comparison, we labeled each instance as correct or incorrect accordingly.

### A.3.1 CALIBRATION METRICS

- **Expected Calibration Error** (ECE; Naeini et al., 2015):

$$\text{ECE} = \sum_{m=1}^{M} \frac{|B_m|}{n} \left| \text{acc}(B_m) - \text{conf}(B_m) \right|$$

  where $B_m$ is the set of predictions in bin $m$, $\text{acc}(B_m)$ is the accuracy, and $\text{conf}(B_m)$ is the average confidence of predictions in that bin. ECE measures how well the model's predicted probabilities are calibrated.

- **Brier Score** (BS; Brier, 1950):

$$\text{BS} = \frac{1}{N} \sum_{i=1}^{N} (f_i - y_i)^2$$

  where $f_i$ is the predicted probability and $y_i$ is the true label. BS combines both the accuracy and confidence of predictions, penalizing overconfident and underconfident predictions.

**Table 4:** Comparison of zero-shot evaluation of calibration baselines on **BioASQ** dataset. Results are averaged over three random seeds.

| Methods | AUROC | ACC | ECE | BS | NLL | %NA |
|---|---|---|---|---|---|---|
| *Base* | - | $27.41 \pm 1.25$ | - | - | - | $26.10 \pm 0.34$ |
| *CT-probe* | $58.11 \pm 1.68$ | $28.19 \pm 0.48$ | $0.3368 \pm 0.03$ | $0.3559 \pm 0.02$ | $1.1195 \pm 0.12$ | $28.05 \pm 3.42$ |
| *CT-LoRA* | $65.74 \pm 0.37$ | $29.05 \pm 0.66$ | $0.3664 \pm 0.03$ | $0.3640 \pm 0.02$ | $1.1729 \pm 0.06$ | $26.83 \pm 2.23$ |
| *Number-LoRA* | $65.40 \pm 2.77$ | $28.84 \pm 0.86$ | $0.2677 \pm 0.00$ | $0.2992 \pm 0.00$ | $0.8220 \pm 0.02$ | $32.43 \pm 0.62$ |
| *Linguistic-LoRA* | $51.72 \pm 1.48$ | $31.02 \pm 0.40$ | $0.2868 \pm 0.01$ | $0.3828 \pm 0.00$ | $1.0311 \pm 0.01$ | $24.28 \pm 0.34$ |
| *CalibRAG* | $71.21 \pm 0.83$ | $35.03 \pm 0.14$ | $0.2500 \pm 0.01$ | $0.2900 \pm 0.01$ | $0.7899 \pm 0.01$ | $27.31 \pm 0.97$ |
| *CalibRAG$^{\dagger}$* | $76.50 \pm 4.98$ | $35.98 \pm 0.38$ | $0.2667 \pm 0.00$ | $0.2779 \pm 0.01$ | $0.7560 \pm 0.04$ | $25.16 \pm 0.42$ |

**Table 5:** Comparison of zero-shot evaluation of calibration baselines on **HotpotQA** dataset. Results are averaged over three random seeds.

| Methods | AUROC | ACC | ECE | BS | NLL | %NA |
|---|---|---|---|---|---|---|
| *Base* | - | $28.47 \pm 3.22$ | - | - | - | $41.82 \pm 7.25$ |
| *CT-probe* | $55.95 \pm 0.75$ | $31.75 \pm 0.33$ | $0.3600 \pm 0.01$ | $0.3773 \pm 0.01$ | $1.2479 \pm 0.01$ | $32.42 \pm 2.68$ |
| *CT-LoRA* | $60.87 \pm 0.37$ | $30.13 \pm 0.75$ | $0.3858 \pm 0.01$ | $0.3842 \pm 0.01$ | $1.4122 \pm 0.05$ | $37.29 \pm 2.61$ |
| *Number-LoRA* | $63.91 \pm 1.97$ | $26.54 \pm 1.03$ | $0.1971 \pm 0.03$ | $0.2643 \pm 0.02$ | $0.7724 \pm 0.08$ | $50.53 \pm 3.59$ |
| *Linguistic-LoRA* | $51.07 \pm 0.62$ | $33.64 \pm 0.10$ | $0.2886 \pm 0.01$ | $0.3100 \pm 0.01$ | $0.9413 \pm 0.04$ | $33.76 \pm 1.53$ |
| *CalibRAG* | $65.47 \pm 0.94$ | $39.28 \pm 0.79$ | $0.2414 \pm 0.03$ | $0.2876 \pm 0.01$ | $0.8276 \pm 0.05$ | $26.85 \pm 2.12$ |
| *CalibRAG$^{\dagger}$* | $68.51 \pm 2.19$ | $40.70 \pm 0.40$ | $0.2392 \pm 0.01$ | $0.2642 \pm 0.01$ | $0.7390 \pm 0.05$ | $25.90 \pm 1.34$ |

- **Negative Log Likelihood (NLL)**:

$$\text{NLL} = -\frac{1}{N} \sum_{i=1}^{N} \log p(y_i \mid x_i)$$

where $p(y_i \mid x_i)$ is the probability assigned to the correct class $y_i$ given input $x_i$. NLL evaluates the model's probabilistic predictions and lower values indicate better calibration.

### A.3.2 HUMAN EVALUATION

We recruited 10 participants to answer 10 questions from each confidence bin, with the survey formatted as shown in Fig. 5. The survey was conducted anonymously, ensuring that no ethical concerns were raised during the process.

**Additional findings of human evaluations.** In Fig. 3a, the 0-20 confidence bin exhibits the lowest agreement between human and user models. Our qualitative analysis revealed that, for the question, "Rex Riot is known for a remix of the Kanye West song from which album?", the model generated the answer, "All of the Lights by Kanye West." with a confidence score of only 0.09. Despite this low confidence, participants trusted the model's output due to the retrieval-augmented guidance that made the response sound convincing. This suggests that "plausible-sounding LLMs" with retrieval-based support can significantly influence people, even when their numerical confidence is low. We leave further exploration of this phenomenon to future research.

## B ADDTIONAL EXPERIMENTS

Table 4, Table 5, Table 6, and Table 7 present the complete results from the primary experiments. For the *Base* model, we utilized a pretrained model, sampling sentences across three different seeds. For the other methods, training was conducted across three random seeds to ensure robust evaluation.

Table 8 and Table 9 present results demonstrating how the existing baselines perform without the application of RAG. It can be observed that RAG generally increases accuracy while also leading

**Table 6:** Comparison of zero-shot evaluation of calibration baselines on **WebQA** dataset. Results are averaged over three random seeds.

| Methods | AUROC | ACC | ECE | BS | NLL | %NA |
|---|---|---|---|---|---|---|
| *Base* | - | $35.84 \pm 0.07$ | - | - | - | $31.16 \pm 1.05$ |
| *CT-probe* | $57.94 \pm 0.67$ | $37.31 \pm 1.85$ | $0.3572 \pm 0.02$ | $0.3724 \pm 0.02$ | $1.2797 \pm 0.17$ | $13.03 \pm 1.48$ |
| *CT-LoRA* | $62.54 \pm 0.69$ | $33.48 \pm 1.07$ | $0.3382 \pm 0.02$ | $0.3507 \pm 0.02$ | $1.0852 \pm 0.01$ | $14.18 \pm 0.73$ |
| *Number-LoRA* | $63.55 \pm 2.27$ | $35.05 \pm 0.10$ | $0.3382 \pm 0.03$ | $0.3372 \pm 0.02$ | $0.9688 \pm 0.04$ | $15.32 \pm 0.67$ |
| *Linguistic-LoRA* | $50.33 \pm 0.20$ | $36.88 \pm 0.42$ | $0.4894 \pm 0.00$ | $0.4809 \pm 0.00$ | $1.3977 \pm 0.01$ | $9.45 \pm 0.79$ |
| *CalibRAG* | $69.58 \pm 0.56$ | $44.32 \pm 0.42$ | $0.3064 \pm 0.04$ | $0.3251 \pm 0.02$ | $0.8919 \pm 0.08$ | $6.65 \pm 0.73$ |
| *CalibRAG*$^{\dagger}$ | $73.23 \pm 1.46$ | $45.03 \pm 0.58$ | $0.3194 \pm 0.03$ | $0.3113 \pm 0.03$ | $0.9050 \pm 0.06$ | $5.43 \pm 0.32$ |

**Table 7:** Comparison of zero-shot evaluation of calibration baselines on **NQ** dataset. Results are averaged over three random seeds.

| Methods | AUROC | ACC | ECE | BS | NLL | %NA |
|---|---|---|---|---|---|---|
| *Base* | - | $36.95 \pm 3.17$ | - | - | - | $30.90 \pm 2.07$ |
| *CT-probe* | $58.15 \pm 1.54$ | $38.53 \pm 2.39$ | $0.3898 \pm 0.03$ | $0.4273 \pm 0.02$ | $1.3313 \pm 0.02$ | $14.93 \pm 2.44$ |
| *CT-LoRA* | $64.08 \pm 1.91$ | $38.89 \pm 0.24$ | $0.3936 \pm 0.01$ | $0.3670 \pm 0.01$ | $1.2574 \pm 0.03$ | $18.05 \pm 0.07$ |
| *Number-LoRA* | $64.83 \pm 2.32$ | $38.40 \pm 0.80$ | $0.2508 \pm 0.01$ | $0.2914 \pm 0.02$ | $0.8707 \pm 0.06$ | $24.23 \pm 2.75$ |
| *Linguistic-LoRA* | $51.37 \pm 1.31$ | $41.44 \pm 0.07$ | $0.4220 \pm 0.01$ | $0.4254 \pm 0.01$ | $1.2433 \pm 0.03$ | $11.86 \pm 0.78$ |
| *CalibRAG* | $66.36 \pm 0.68$ | $48.29 \pm 0.59$ | $0.2596 \pm 0.02$ | $0.2994 \pm 0.02$ | $0.8490 \pm 0.01$ | $8.39 \pm 0.54$ |
| *CalibRAG*$^{\dagger}$ | $69.40 \pm 2.90$ | $49.10 \pm 0.17$ | $0.2625 \pm 0.00$ | $0.2876 \pm 0.02$ | $0.8150 \pm 0.02$ | $8.39 \pm 0.38$ |

**Table 8:** Additional performance comparison of baselines with and without RAG. When applying RAG on the HotpotQA dataset, we observe that the overall accuracy improves, but the calibration error increases.

| | Methods | AUROC | ACC | ECE | BS | NLL | %NA |
|---|---|---|---|---|---|---|---|
| **No RAG** | *CT-probe* | 61.32 | 15.46 | 0.4224 | 0.4208 | 1.7531 | 65.56 |
| | *CT-LoRA* | 58.27 | 17.93 | 0.3394 | 0.3623 | 1.0450 | 46.48 |
| | *Number-LoRA* | 62.39 | 25.86 | 0.1887 | 0.1552 | 0.6846 | 50.22 |
| | *Linguistic-LoRA* | 61.25 | 26.32 | 0.2614 | 0.2353 | 0.7430 | 49.28 |
| **RAG** | *CT-probe* | 56.31 | 31.43 | 0.3583 | 0.3846 | 1.2633 | 36.20 |
| | *CT-LoRA* | 60.80 | 30.25 | 0.3979 | 0.3984 | 1.3415 | 38.58 |
| | *Number-LoRA* | 61.14 | 25.51 | 0.2366 | 0.2935 | 0.8927 | 54.65 |
| | *Linguistic-LoRA* | 51.70 | 33.54 | 0.2787 | 0.3279 | 0.8959 | 35.30 |

to a rise in calibration error. And these results empirically validate that the LLM model $\mathcal{M}$ places strong trust in the retrieved documents, leading to overconfidence in the generated guidance. As a result, while accuracy increases, the calibration performance significantly decreases. Therefore, these results suggest that additional calibration adjustments are necessary when applying RAG to ensure balanced performance between accuracy and calibration. CalibRAG demonstrates both high accuracy and improved calibration metrics in such scenarios.

**Analysis of $\epsilon$** In our experiments, $\epsilon$ was set as a balanced choice to manage the trade-off between accuracy and calibration error. As shown in Table 10, increasing $\epsilon$ results in retrieving a larger number of new queries, incorporating more relevant information, and thereby improving accuracy. However, this increase can potentially lead to higher calibration errors. Specifically, while better retrieval enhanced prediction accuracy, the confidence scores for these predictions only increased marginally. This mismatch between improved accuracy and relatively low confidence resulted in underconfident predictions, which contributed to a slight increase in calibration error.

**Table 9:** Additional performance comparison of baselines with and without RAG. When applying RAG on the NatrualQA dataset, we observe that the overall accuracy improves, but the calibration error increases.

|        | Methods        | AUROC | ACC   | ECE    | BS     | NLL    | %NA   |
|--------|----------------|-------|-------|--------|--------|--------|-------|
| **No RAG** | *CT-probe*        | 51.08 | 33.10 | 0.3496 | 0.3790 | 1.0235 | 24.68 |
|        | *CT-LoRA*         | 55.51 | 39.00 | 0.3021 | 0.3487 | 1.0326 | 18.15 |
|        | *Number-LoRA*     | 63.64 | 33.76 | 0.1085 | 0.1610 | 0.7819 | 26.35 |
|        | *Linguistic-LoRA* | 50.74 | 42.83 | 0.4497 | 0.4486 | 1.3080 | 13.74 |
| **RAG**    | *CT-probe*        | 58.31 | 35.23 | 0.4194 | 0.4233 | 1.3196 | 15.93 |
|        | *CT-LoRA*         | 65.87 | 39.01 | 0.3762 | 0.3743 | 1.2092 | 18.12 |
|        | *Number-LoRA*     | 61.55 | 37.27 | 0.2500 | 0.3010 | 0.8575 | 28.04 |
|        | *Linguistic-LoRA* | 52.67 | 41.37 | 0.4124 | 0.4154 | 1.2130 | 12.64 |

**Table 10:** Effect of Threshold Selection on Performance. Experiments on the BioASQ dataset show how increasing $\epsilon$ affects accuracy and calibration metrics.

| $\epsilon$ | AUROC | ACC | ECE | BS | NLL | %NA |
|-----|-------|-----|-----|-----|-----|-----|
| 0.0 | $71.21 \pm 0.83$ | $35.03 \pm 0.14$ | $0.2500 \pm 0.01$ | $0.2900 \pm 0.01$ | $0.7899 \pm 0.01$ | $27.31 \pm 0.97$ |
| 0.4 | $76.15 \pm 1.50$ | $35.05 \pm 0.25$ | $0.2608 \pm 0.00$ | $0.2830 \pm 0.00$ | $0.7703 \pm 0.03$ | $26.57 \pm 0.80$ |
| 0.5 | $76.50 \pm 4.98$ | $35.98 \pm 0.38$ | $0.2667 \pm 0.00$ | $0.2779 \pm 0.01$ | $0.7560 \pm 0.04$ | $25.16 \pm 0.42$ |
| 0.6 | $77.20 \pm 4.10$ | $36.50 \pm 0.45$ | $0.2707 \pm 0.00$ | $0.2800 \pm 0.01$ | $0.7620 \pm 0.03$ | $24.98 \pm 0.50$ |

**Table 11:** Evaluation results on TREC-COVID and SciFact datasets, a subset of the BEIR benchmark. The evaluation metric is Normalized Discounted Cumulative Gain (NDCG@K).

| Model         | Dataset     | NDCG@5 | NDCG@10 |
|---------------|-------------|--------|---------|
| Cross-Encoder | TREC-COVID  | 0.7655 | 0.7576  |
|               | SciFact     | 0.6668 | 0.6914  |
| CalibRAG      | TREC-COVID  | 0.7863 | 0.7660  |
|               | SciFact     | 0.6872 | 0.7114  |

To assess the impact of different $\epsilon$ values on model performance, we conducted experiments on the BioASQ dataset. Based on these observations, we selected $\epsilon = 0.5$ as a reasonable compromise to balance accuracy improvements with calibration reliability.

**Evaluation on BEIR Benchmark**    To provide a more comprehensive evaluation, we conducted experiments using two datasets from the BEIR benchmark: SciFact and TREC-COVID. These evaluations aim to validate the effectiveness of CalibRAG beyond its primary focus on well-calibrated decision-making, which predicts the probability of a correct decision when a user relies on the generated guidance to solve a given problem. While CalibRAG is not specifically designed as a reranking method to optimize retrieval performance, it inherently supports both calibration and retrieval.

For the experiments, we followed the standard retrieval pipeline, retrieving documents using BM25 and reranking the top-100 results. We compared CalibRAG with the Cross-Encoder baseline, and the results, presented in Table 11, demonstrate that CalibRAG consistently outperforms the Cross-Encoder. These findings validate that CalibRAG not only enables well-calibrated decision-making but also enhances retrieval performance, reinforcing its utility in relevant scenarios.

**Analysis of Verbalized Confidence Representations**    CalibRAG does not rely on linguistic or numerical confidence in its primary approach. Instead, it provides confidence scores based on probability predictions generated by the forecasting function. Verbalized confidence, however, was used

**Table 12:** Results of Verbalized Confidence Fine-Tune Evaluation on the MMLU Dataset using Llama-3-8B. Evaluation metrics are ACC and ECE.

| Case | ACC | ECE |
|---|---|---|
| Continuous-Number | 43.63 | 0.3190 |
| Discrete-Number | 44.96 | 0.1605 |
| Linguistic | 45.03 | 0.1585 |

**Table 13:** Comparison of Agreement Rates with Human Decisions Across Confidence Ranges.

| Confidence Range (%) | Agreement Rate (With Prompt) | Agreement Rate (Without Prompt) |
|---|---|---|
| 0-20 | 70.00% | 30.00% |
| 20-40 | 80.00% | 70.00% |
| 40-60 | 60.00% | 40.00% |
| 60-80 | 96.67% | 96.67% |
| 80-100 | 100.00% | 100.00% |
| **Average** | **81.33%** | **67.33%** |

as a baseline in the comparative models. Verbalized confidence is typically expressed as a continuous number within the range [0, 100] Tian et al. (2023); Xiong et al. (2023), but LLMs often struggle to interpret these numerical values precisely.

To address this limitation, alternative representations were explored in the baselines: (1) linguistic expressions (e.g., "likely"), and (2) discrete numerical values ranging from 0 to 10. These approaches were termed Linguistic and Number, respectively, with detailed prompt designs provided in Appendix E.

To further analyze verbalized confidence, we conducted experiments on the MMLU dataset using the Llama-3-8B model. We evaluated the effectiveness of three confidence representations: continuous number, discrete number, and linguistic. As shown in Table 12, both discrete number and linguistic representations outperformed the continuous number baseline. Linguistic confidence, in particular, addressed the limitations of the model's understanding of numerical relationships and improved calibration.

**Comparison of Agreement Rates with Human Decisions** We acknowledge that an LLM cannot fully replicate human behavior with 100% accuracy. However, conducting evaluations with multiple human annotators for all data would involve substantial costs, and our annotator pool was limited in size and included some outliers. Furthermore, when the LLM made decisions independently, without prompts designed to simulate human decision-making, a significant gap was observed between human and surrogate model decisions.

To address this, we employed prompts to minimize this gap. Table 13 shows a comparison of agreement rates with human decisions across different confidence ranges. The results demonstrate that incorporating prompts significantly improves the agreement rates, especially in lower confidence ranges, reducing the gap between human and surrogate model decisions.

**Comparison with RAG Robustness Methods** There are many methods like CRAG (Yan et al., 2024), Self-RAG (Asai et al., 2023), and RetRobust (Yoran et al., 2023) designed to improve the robustness of RAG systems. However, these approaches are fundamentally different from CalibRAG. While CRAG focuses on evaluating the correctness of documents based on relevance, Self-RAG measures utility as the perceived informativeness of answers, and RetRobust learns whether a query can be inferred from a document. In contrast, CalibRAG explicitly models the accuracy of user decisions and aims to provide reliable calibration by maximizing proper scoring rules.

**Table 14:** Comparison of CalibRAG and Self-RAG in Zero-Shot Decision Calibration.

| Method | Dataset | ACC | AUROC | ECE | NLL | BS | %NA |
|--------|---------|-----|-------|-----|-----|-----|-----|
| Self-RAG | BioASQ | $30.15 \pm 0.07$ | $50.00 \pm 0.04$ | $0.6932 \pm 0.07$ | $23.94 \pm 0.08$ | $0.6932 \pm 0.09$ | $1.72 \pm 0.09$ |
| | HotpotQA | $33.92 \pm 0.10$ | $50.11 \pm 0.01$ | $0.6507 \pm 0.00$ | $22.43 \pm 0.09$ | $0.6505 \pm 0.04$ | $2.77 \pm 0.08$ |
| | WebQA | $38.83 \pm 0.02$ | $50.00 \pm 0.09$ | $0.6104 \pm 0.03$ | $21.07 \pm 0.04$ | $0.6104 \pm 0.11$ | $3.00 \pm 0.02$ |
| | NQ | $34.97 \pm 0.09$ | $50.09 \pm 0.04$ | $0.6471 \pm 0.10$ | $22.33 \pm 0.07$ | $0.6469 \pm 0.05$ | $8.00 \pm 0.05$ |
| CalibRAG | BioASQ | $35.03 \pm 0.14$ | $71.21 \pm 0.83$ | $0.2500 \pm 0.01$ | $0.7899 \pm 0.01$ | $0.2900 \pm 0.01$ | $27.31 \pm 0.97$ |
| | HotpotQA | $39.28 \pm 0.79$ | $65.47 \pm 0.94$ | $0.2414 \pm 0.03$ | $0.8276 \pm 0.05$ | $0.2876 \pm 0.01$ | $26.85 \pm 2.12$ |
| | WebQA | $44.32 \pm 0.42$ | $69.58 \pm 0.56$ | $0.3064 \pm 0.04$ | $0.8919 \pm 0.08$ | $0.3251 \pm 0.02$ | $6.65 \pm 0.73$ |
| | NQ | $48.29 \pm 0.59$ | $66.36 \pm 0.68$ | $0.2596 \pm 0.02$ | $0.8490 \pm 0.01$ | $0.2994 \pm 0.02$ | $8.39 \pm 0.54$ |

**Table 15:** Examples of Query Reformulation

| Case | Original Query | Reformulated Query |
|------|----------------|--------------------|
| 1 | Write a paragraph about the effect of TRH on myocardial contractility. | Write a paragraph about the effect of Thyrotropin-Releasing Hormone (TRH) on myocardial contractility. |
| 2 | Write a paragraph about the clinical trials for off-label drugs in neonates as cited in the literature. | Write a paragraph about clinical trials for off-label drug use in neonates as reported in the medical literature. |
| 3 | Write a paragraph about the current representatives from Colorado. | Write a paragraph about the current representatives from the state of "Colorado" in the United States. |
| 4 | Write a paragraph about the current minister of local government in Zimbabwe and their role within the government. | Write a paragraph about the current Minister of Local Government and Public Works in Zimbabwe and their role within the government. |

To further investigate these differences, we conducted experiments using the same settings for all methods. As shown in Table 14, CalibRAG demonstrates significantly lower ECE compared to SelfRAG, highlighting its effectiveness in providing well-calibrated decision guidance.

## C EXAMPLES OF QUERY REFORMULATIONS

In CalibRAG, the initial query is generated to simulate how a human decision-maker might pose a simple query based on the input. For example, a decision-maker faced with a problem such as "Is a tomato a fruit or a vegetable?" might craft a straightforward query like "Classification of tomatoes" to query a language model. Using this setup, we employed an LLM generator to create simple yet relevant queries and retrieved documents based on these queries. If the retrieved documents were insufficiently informative, the query was reformulated in Stage 3. This reformulation emphasized key terms to refine the query and improve the quality of retrieved documents. The specific prompt used for this process is detailed in Appendix F.

To help readers understand the transformation from the initial query to its reformulated version, Table 15 provides examples illustrating how queries evolve during the refinement process, offering practical insights into the mechanism.

## D DATA EXAMPLES

Fig. 6 shows the top 20 examples of queries and their corresponding labels. The full set of data examples will be released upon publication of the paper. Fig. 6 shows that ranking of the retrieved documents does not correlate with the accuracy of the user decision. As seen in this example, the top-ranked document is not helpful for the user model in decision-making, whereas the second-ranked document provides information that can lead the user model to make a correct decision. This illustrates the importance of CalibRAG's forecasting function $f$ in effectively modeling the

probability that a decision made using document $d$ is correct, emphasizing the need for reranking documents based on this modeling.

Open-ended query: Write a paragraph about the founding year of Apple Computer.

line became a sales smash, moving about one million units each year. It also helped re-introduce Apple to the media and public, and announced the company's new emphasis on the design and aesthetics of its products. In 1999, Apple introduced the Power Mac G4, which utilized the Motorola-made PowerPC 7400 containing a 128-bit instruction unit known as AltiVec, its flagship processor line. Also that year, Apple unveiled the iBook, its first consumer-oriented laptop that was also the first Macintosh to support the use of Wireless LAN via the optional AirPort card that was based on the 802.11b standard; it helped **(False)**

Xcode. Its online services include the iTunes Store, the iOS App Store and Mac App Store, Apple Music, and iCloud. Apple was founded by Steve Jobs, Steve Wozniak, and Ronald Wayne in April 1976 to develop and sell Wozniak's Apple I personal computer. It was incorporated as Apple Computer, Inc., in January 1977, and sales of its computers, including the Apple II, grew quickly. Within a few years, Jobs and Wozniak had hired a staff of computer designers and had a production line. Apple went public in 1980 to instant financial success. Over the next few years, Apple shipped new **(True)**

had told him about it and had said he needed the money, Wozniak would have given it to him. In 1975, Wozniak began designing and developing the computer that would eventually make him famous, the Apple I. On June 29 of that year, he tested his first working prototype, displaying a few letters and running sample programs. It was the first time in history that a character displayed on a TV screen was generated by a home computer. With the Apple I, he and Jobs were largely working to impress other members of the Palo Alto-based Homebrew Computer Club, a **(False)**

at the Homebrew Computer Club. Apple I was sold as a motherboard (with CPU, RAM, and basic textual-video chips), which was less than what is now considered a complete personal computer. The Apple I went on sale in July 1976 and was market-priced at $666.66 ($ in dollars, adjusted for inflation). Apple Computer, Inc. was incorporated on January 3, 1977, without Wayne, who left and sold his share of the company back to Jobs and Wozniak for $800 only a couple weeks after co-founding Apple. Multimillionaire Mike Markkula provided essential business expertise and funding of $250,000 during the incorporation of **(True)**

.
.
.

. Apple. During the first five years of operations revenues grew exponentially, doubling about every four months. Between September 1977 and September 1980, yearly sales grew from $775,000 to $118million, an average annual growth rate of 533%. The Apple II, also invented by Wozniak, was introduced on April 16, 1977, at the first West Coast Computer Faire. It differed from its major rivals, the TRS-80 and Commodore PET, because of its character cell-based color graphics and open architecture. While early Apple II models used ordinary cassette tapes as storage devices, they were superseded by the introduction of a -inch floppy **(True)**

of "Kilobaud Microcomputing", publisher Wayne Green stated that "the best consumer ads I've seen have been those by Apple. They are attention-getting, and they must be prompting sale." In August, the "Financial Times" reported that On December 12, 1980, Apple launched the Initial Public Offering of its stock to the investing public. When Apple went public, it generated more capital than any IPO since Ford Motor Company in 1956 and instantly created more millionaires (about 300) than any company in history. Several venture capitalists cashed out, reaping billions in long-term capital gains. In January 1981, Apple held its first shareholders . **(False)**

with no programming language built-in. This presented a problem to Apple: the Mac was due to be launched in 1983 (originally), with a new user interface paradigm, but no third-party software would be available for it, nor could users easily write their own. Users would end up with a computer that did nothing. In order to fill this void, several members of the Mac team took it upon themselves to write simple applications to fill these roles until third-party developers published more full-fledged software. The result was MacWrite and MacPaint, which shipped free with every Macintosh from 1984 to 1986. **(False)**

idea was to design technology based on a profile that included diskless computers, commonly coded applications using languages such as Java, and interface with the internet using common software such as Netscape Navigator. In May 1996, Apple became a partner in the network computing effort, and used the Apple Pippin platform as its implementation. On July 9, 1997, Gil Amelio was ousted as CEO of Apple by the board of directors. Steve Jobs stepped in as the interim CEO ("iCEO", as he was referred to), to begin a critical restructuring of the company's product line. He would eventually become CEO **(False)**

Apple Writer Apple Writer is a word processor for the Apple II family of personal computers. It was created by Paul Lutus and published in 1979 by Apple Computer. Paul Lutus wrote "Apple Writer" alone in a small cottage he built himself atop a hill in the woods of Oregon, connected to the electricity grid via of cable strung in trees. The original 1979 version of "Apple Writer" ran from a 13-sector DOS 3.2 diskette and supported 40-column text display. It displayed text entirely in uppercase, but case could be toggled by pressing the ESC key; characters that the user **(False)**

**Figure 6:** Top-20 retrieved document exampels.

**Original question:** When was the American lawyer, lobbyist and political consultant who was a senior member of the presidential campaign of Donald Trump born?
**Open-ended query:** Write a paragraph about the American lawyer, lobbyist, and political consultant who was a senior member of Donald Trump's presidential campaign, including details about when he was born.

**Answer:** April 1, 1949

**Original Retrieval Model's Top-1 document** (This context set the LM model's confidence to 'Certain'):
… Page testified that he did not "directly" express support for lifting the sanctions during the meeting with Baranov, but that he might have mentioned the proposed Rosneft transaction. Carter Page Carter William Page (born June 3, 1971) is an American petroleum industry consultant and a former foreign-policy adviser to Donald Trump during his 2016 Presidential election campaign.
**User decision: June 3, 1971.**

**CalibRAG's Top-1 document** (This context set the CalibRAG's confidence to 83.93):
Paul Manafort Paul John Manafort Jr. (born April 1, 1949) is an American lobbyist, political consultant, lawyer, and felon. A Republican, he joined Donald Trump's presidential campaign team in March 2016, and was campaign chairman from June to August 2016. Manafort was an adviser to the U.S. presidential campaigns of Republicans Gerald Ford, Ronald Reagan, George H. W. Bush, and Bob Dole. In 1980, he co-founded the Washington, D.C.-based lobbying firm Black, Manafort & Stone, along with principals Charles R. Black Jr., and Roger J. Stone, joined by Peter G. Kelly in 1984. Manafort often lobbied on behalf of …
**User decision: April 1, 1949.**

**Figure 7: CalibRAG vs Linguistic-LoRA.** In the case of CalibRAG, a document about the person in question is retrieved with a confidence level of $83.93\%$. In contrast, the document retrieved by the base retrieval model is related to Donald Trump but does not match the specific person in the query. Nevertheless, the Linguistic-LoRA model trust the document confidently.

**Original question:**
Where did the allies go after north Africa?
**Open-ended query:**
Write a paragraph about the next steps taken by the Allies following their involvement in North Africa.

**Answer:** Italy

**Original Retrieval Model's Top-1 document** (This context set the LM model's confidence to 6/10):
… Abteilung 350 procedures, subsequently the German High Command was completely taken by surprise by the bulk of the Allies Mediterranean theatre buildup and subsequent landing in French North Africa on 7 November 1942, that was known by the allies as Operation Torch. Not until the Tunisian Campaign was well under way did W-Leit 2 at Taormina pull itself together and contribute to the general effort by intercepting air support messages. It is worth noting that General Albert Praun, who was chief Signals of the Wehrmacht thought that the Allied Invasion of French North Africa was not detected due to radio…
**User decision: Based on the context provided, the Allies advanced towards Tunisia after North Africa.**

**CalibRAG's Top-1 document** (This context set the CalibRAG's confidence to 77.98):
1943, at the Casablanca Conference, the Allies agreed to demand Axis unconditional surrender, a war aim which implied the physical occupation of Germany with land forces. While building up strength for an invasion of continental Europe, the Allies pursued an indirect strategy by invading Europe from the South. After defeating Axis forces in North Africa (the invasion of French North-Africa), Sicily and southern Italy were invaded, leading to the defeat of Fascist Italy. Churchill especially favoured a Southern strategy, aiming to attack the "soft underbelly" of Axis Europe through Italy, Greece and the Balkans in a strategy similar to the> Question: Write a paragraph about the next steps taken by the Allies following their involvement in North Africa.
**User decision: The Allies went to Sicily.**

**Figure 8: CalibRAG vs Number-LoRA.** In the case of CalibRAG, an accurate document about the location following North Africa was retrieved, allowing the user model to make a correct decision. In contrast, the base retrieval model brought a different document. Nevertheless, Number-LoRA model set this context with a confidence level of 6 out of 10, leading the user to ultimately make an incorrect decision.

## E  QUALITATIVE EXAMPLES

Here, we present additional qualitative examples for comparison with other baselines. In Fig. 7, Fig. 8, Fig. 9, and Fig. 10, the examples demonstrate that while the baselines retrieve documents that provide incorrect answers to the queries, they still assign high confidence to the retrieved documents. In contrast, CalibRAG effectively reranks and retrieves documents that are highly relevant to the decision problem $x$, allowing us to confirm that the guidance generated from these retrieved documents is well-predicted to be helpful for decision-making. Additionally, we can confirm that when the document with the highest rank does not aid in decision-making for $x$, CalibRAG successfully assigns a lower confidence level, helping to prevent the user from over-relying on the guidance.

**Original question:**
Who is the first president to be impeached?
**Open-ended query:**
Write a paragraph about the first president to be impeached.

**Answer:** Andrew Johnson

**Original Retrieval Model's Top-1 document** (This context set the LM model's confidence to 9/10):
Constitution and the whole history of our republic demand that we make up our minds." The Judiciary Committee, with six Republicans joining the Democratic majority, passed three of the five proposed articles of impeachment. On July 27, 1974, the Committee voted 27–11 to recommend the first article of impeachment against the president: obstruction of justice. The Committee then recommended the second article, abuse of power, on July 29, 1974. The next day, on July 30, 1974, the Committee recommended the third article: contempt of Congress. Article I alleged in part: On June 17, 1972, and prior thereto, agents of the…
**User decision: Richard Nixon.**

**CalibRAG's Top-1 document** (This context set the CalibRAG's confidence to 92.32):
… justice following allegations that he committed perjury and obstructed justice to conceal an affair he had with Monica Lewinsky, a 22-year old White House intern. Clinton was acquitted by the Senate in 1999 and completed his term in office. He is only the second U.S. president to ever be impeached, the first being Andrew Johnson. During the last three years of Clinton's presidency, the Congressional Budget Office reported a budget surplus, the first such surplus since 1969. In foreign policy, Clinton ordered U.S. military intervention in the Bosnian and Kosovo wars, signed the Iraq Liberation Act in opposition to Saddam …
**User decision: Andrew Johnson.**

**Figure 9: CalibRAG vs Number-LoRA.** The base retrieval model focused solely on the word 'impeached' and retrieved a related document, missing the context of 'first.' Despite this, Number-LoRA model set a confidence level of 9 out of 10, causing the user to make an incorrect decision. In contrast, CalibRAG retrieved an accurate document that, while not explicitly containing 'first impeached,' included the phrase 'first being.' It set a confidence level of 92.32%, allowing the user to arrive at the correct answer.

**Original question:**
what year is it for the Jewish calendar?
**Open-ended query:**
Write a paragraph about the current year in the Jewish calendar.

**Answer:** AM 5778

**Original Retrieval Model's Top-1 document** (This context set the LM model's confidence to 96.83):
…of two or three Hebrew letters. See Four gates. In Hebrew there are two common ways of writing the year number: with the thousands, called ("major era"), and without the thousands, called ("minor era"). Thus, the current year is written as ' () using the "major era" and ' (%1000) using the "minor era". In 1178 CE, Maimonides wrote in the "Mishneh Torah", "Sanctification of the Moon" (11.16), that he had chosen the epoch from which calculations of all dates should be as "the third day of Nisan in this present year ... which is the year 4938 of the…
**User decision: According to the provided context, the current year in the Hebrew calendar is written as 5784 (with the thousands) and 2444 (without the thousands). Given the model's high confidence level.**

**CalibRAG's Top-1 document** (This context set the CalibRAG's confidence to 20.95):
…year; and about every 231 years it will fall a day behind the mean Gregorian calendar year. The era used since the Middle Ages is the epoch (Latin for "in the year of the world"; , "from the creation of the world"). As with (A.D. or AD), the words or abbreviation for (A.M. or AM) for the era should properly "precede" the date rather than follow it. AM began at sunset on and will end at sunset on . The Jewish day is of no fixed length. The Jewish day is modeled on the reference to "...there was evening and…
**User decision: I'm not able to provide the current year in the Jewish calendar based on the given context and the model's confidence level. The model's confidence level is 20.95, which is below.**

**Figure 10: CalibRAG vs CT-LoRA.** In the case of CalibRAG, the top-20 confidence score is 20.95 for incorrect information, causing the user to hesitate in making a decision. However, with the CT-LoRA model, incorrect information is assigned a confidence score of 96.83, leading the user to make an incorrect decision.

# F  PROMPT EXAMPLES

```
f"""The task is to answer questions based on a context generated by a
    language model in response to a question about relevant information,
    along with the model's confidence level in the provided answer.

Follow the behavior instructions below:
1. Overconfidence: You feel that the answers provided are very accurate.
    Trust the provided answers with confidence.
2. Avoidance of Uncertainty: If any uncertainty is expressed in the
    answer, do not trust the answer and seek other information.
3. Consistent Trust: If the provided answer shows clear confidence or
    lacks any explicit uncertainty, accept the answer as it is.
4. Trust in Incorrect Information: You believe that answers provided with
     confidence are correct, even if they are actually incorrect.

If you cannot provide an answer or if you do not trust the information,
    respond with 'no answer'.

Additionally, consider the language model's confidence level when
    assessing the answer.

Context: {context}
Question: {question}
Model Confidence: {confidence}
Answer:"""
```

**Table 1: Human decision prompt.** This prompt is crafted to force the user model to imitate according to the human evaluation results of Zhou et al. (2024). This prompt encourages the user model $U$ to over-rely on the guidance provided by the LLM.

In this section, we present prompt examples used during training and inference. Table 1 shows the prompt that encourages the user model $U$ to act like a human decision-maker, leading it to over-rely on the guidance provided by the LLM. Table 2 displays the prompt that generates the open-ended query $q$ from the decision task $x$. Table 3 presents the prompt that induces the generation of guidance $z$ from $M$ based on the retrieved document $d$. Table 4 is used when grading the user model $U$'s decision against the true answer using $\mathcal{G}$. Table 5, Table 6, and Table 7 are prompts used to instruct $\mathcal{M}$ to generate confidence in terms of linguistic or numerical calibration. Lastly, Table 8 is the prompt used during **Stage 3** of the inference process.

```
f"""You are an automated assistant tasked with rephrasing specific
    questions into open-ended queries to encourage detailed exploration
    and discussion of the key topics mentioned.

Your goal is to prompt someone to write a paragraph exploring the topic
    without directly revealing the answer.

You will be given an original question, labeled as 'Question 1.' Your
    task is to rephrase this into a new question, labeled as 'Question
    2.' This new question should encourage someone to provide a
    comprehensive exploration of the key topic from the original question
    .

Examples for Guidance:

Example 1:
Question 1: Which sea creature is the world's largest invertebrate?
Question 2: Write a paragraph about the world's largest invertebrate.

Example 2:
...

Example 3:
Question 1: In which century was the printing press established in
    Britain?
Question 2: Write a paragraph about the century in which the printing
    press was established in Britain.

Example 4:
Question 1: What type of creature is Chewbacca?
Question 2: Write a paragraph about the type of creature that Chewbacca
    is.

Now, please rephrase the following question:
Question 1: {question}
Question 2:"""
```

Table 2: **Prompt that generates open-ended query** $q$ **from the decision task** $x$**.** This prompt was first suggested by Band et al. (2024), and we have modified part of the proposed prompt for our use here. We use this prompt as an input when generating the query $q$ based on the decision task $x$.

```
f"""You are an expert who responds with concise, correct answers.
    Directly state the answer without phrases like 'the correct answer is
    .'
If you cannot provide an answer or if you do not trust the information,
    respond with 'no answer.'
Given the provided context, answer the question based on that context.

Context: {title};{context}
Question: {query}
Answer:"""
```

Table 3: **Guidance** $z$ **generation prompt.** This prompt guides the language model to provide direct, concise guidance $z$ based on a given retrieved document $d$, avoiding unnecessary phrases. It emphasizes providing an answer or stating 'no answer' if information is insufficient.

```
f"""The problem is: {question}

The correct answer for this problem is: {ground-truth}

A student submitted the answer: {prediction}

The student's answer must be correct and specific but not overcomplete (
    for example, if they provide two different answers, they did not get
    the question right). However, small differences in formatting should
    not be penalized (for example, 'New York City' is equivalent to 'NYC
    ').

Did the student provide an equivalent answer to the ground truth? Please
    answer yes or no without any explanation:"""
```

**Table 4: Evaluation prompt.** This prompt was first suggested by Kapoor et al. (2024), and we have modified part of the proposed prompt for our use here. This prompt is used when $\mathcal{G}$ compares the user's decision with the true answer.

```
f"""Is the proposed answer correct?
Choices:
(i): no
(ii): yes
Answer:"""
```

**Table 5: Calibration tuning prompt.** This prompt was first suggested by Kapoor et al. (2024). It poses a straightforward question to verify the correctness of a proposed answer with binary choices for evaluation. We used this prompt when training our baselines.

```
f"""Provide the certainty level of answer using the given 11 certainty
    levels. Give ONLY your certainty level, no other words or explanation
    .

Certainty Levels: Unlikely, Doubtful, Uncertain, Ambiguous, Probable,
    Likely, Possible, Specified, Confirmed, Certain, Inevitable.

For example: Certainty: <ONLY the certainty level that Answer is correct,
    without any extra commentary whatsoever; just the certainty level!>

Certainty:"""
```

**Table 6: Linguistic calibration prompt.** This prompt requires the model to evaluate the certainty of an answer using a predefined set of linguistic levels of certainty. We used this prompt for our baselines that utilize linguistic calibration.

```
f"""Provide the certainty level of answer using the given 11 certainty
    levels. Give ONLY your certainty level, no other words or explanation
    .

Certainty Levels: 0, 1, 2, 3, 4, 5, 6, 7, 8, 9, 10.

For example: Certainty: <ONLY the certainty level that Answer is correct,
    without any extra commentary whatsoever; just the number!>

Certainty:"""
```

**Table 7: Number calibration prompt.** This prompt is similar to the linguistic calibration prompt but uses numerical certainty levels (from 0 to 10) to rate the confidence in the answer provided. We used this prompt for our baselines that utilize number calibration.

```
f"""You are a language model assistant who specializes in improving
    queries for document search systems. Your task is to highlight and
    clarify the important parts of a given query to make it more precise
    and help retrieve relevant documents.

Please take the original search query below and rewrite it by emphasizing
    the important words. Do not add any new information not included in
    the original query.

Original Retrieval Query: {query}

Please generate the new retrieval query without any explanation:"""
```

**Table 8: Query regeneration prompt.** This prompt assists in rewriting search queries to enhance precision and relevance for document retrieval, emphasizing the crucial elements without adding extraneous information.

