# OpenReview forum: "Calibrated Decision-Making through Large Language Model-Assisted Retrieval"
_ICLR.cc/2025/Conference — Submitted to ICLR 2025_

### Official Review · Reviewer_46aU · 2024-10-27

**Soundness:** 2
**Presentation:** 1
**Contribution:** 2
**Rating:** 3
**Confidence:** 3

**Summary:**

The paper proposes CalibRAG, which extends Band et al. 2024 to the RAG use case, where LLM generates calibrated generations for better decision making. The main component is to train a forecasting function f that provides a confidence estimation of the query and a retrieved document.  Experiments are conducted on 4 existing QA datsets with baselines that were not specifically designed for RAG or reranking methods without considering calibration.

**Strengths:**

1. The topic of confidence estimation in RAG systems (for decision making) is an interesting extension of existing work.

2. In experiments, baselines include both calibration methods and reranking methods, which are two good perspectives to study.

**Weaknesses:**

1. The reviewer feels the writing of the paper is not clear.  The paper does not clearly describe the use case (LLM-assisted decision making). More specifically, almost half of abstract and intro are about very basic concepts about LLM and RAG. The discussion of calibration is also very general and does not talk about anything specific in this work. By the end of abstract and intro, the reviewer is not clear what the paper wants to do. Section 2.1 then immediately starts with Band et al. 2024 and the notations become confusing since the problem setup is unclear. The authors seem to assume the readers know Band et al. 2024 very well. In fact, the reviewer has to read that paper to understand what the problem setup is. The authors are suggested to refer to the paper on how to set up the problem.

2. The novelty and technical contribution of this work looks borderline. More specifically, extending to RAG is meaningful, as indicated above. The work heavily builds upon Band et al. 2024. The arguments of the weaknesses of Band et al. 2024 look reluctant expect for the RAG part. To the reviewer, the real major contribution of the work is to the extension to RAG use case, which is not very technically deep (but again, still meaningful).

3. Many technical details seem to be missing. For example, there's pretty much no details about the cross-encoder baseline. The LLM-rerank baseline is also a very sensitive method. Without more details. the experimental results are not very trustworthy to the reviewer.

**Questions:**

See above.

---

> ### Author Response · Authors · 2024-11-20
>
> >**[W1] The paper lacks clarity, especially in defining the LLM-assisted decision-making use case.**
>
> Thank you for raising your concerns. We hope our explanation helps enhance your understanding. Before reviewing our response, we kindly ask that you also consider the **General Response**, where we provide additional details on the objectives and necessity of our study.
>
> 1. **Use Cases for LLM-Assisted Decision-Making**
>    Our study focuses on scenarios where users rely on LLMs to support decision-making. For instance, in a medical diagnosis context, users may ask the LLM questions about their symptoms and make decisions based on their responses and the confidence expressed within them. For example, a user might ask, "Which hospital should I visit?" Zhou et al. (2023) observed that people are highly influenced by the **confidence expressed** by LLMs. When users overly rely on such outputs, there is a risk of suboptimal decisions if the LLM exhibits overconfidence (Lines 35-38).
>
> 2. **Our Approach**
>    Our approach to decision calibration aims to align the confidence expressed by the LLM with the actual uncertainty of the information it provides. For instance, when the LLM assigns lower confidence to uncertain information, users are more likely to make cautious decisions. This can be achieved using **proper scoring rules**, such as maximizing the **log-likelihood** of the probability of correctness (Lines 69-73). In contrast to Band et al.'s **linguistic calibration**, which explicitly adjusts verbalized confidence during training and incurs significant computational costs due to **next token prediction**, our approach minimizes computational overhead while addressing the same goal. Moreover, our findings reveal that Retrieval-Augmented Generation (RAG) can exacerbate overconfidence issues. Therefore, we propose a novel framework designed to mitigate these challenges effectively and efficiently (Lines 69-73).
>
> If there are still unclear points, please feel free to ask further questions. We are happy to provide additional clarification to address your concerns.
>
>
> >**[w2] The novelty and technical contribution are marginal, with the primary contribution being the extension to the RAG use case, which lacks depth.**
>
> We understand your concerns, but we would like to repeat the **novelty** of our research.
>
> The motivation for addressing and resolving the issue of decision calibration in RAG becomes clear when compared to the approach by Band et al. (2024). Their method requires fine-tuning three separate models across four stages and uses PPO for calibration, which can lead to instability during training. Additionally, existing **calibration methods that do not account for RAG**, as shown in Figure 1-(b), face difficulties integrating with RAG and carry the risk of LLMs being overly confident in the retrieved context. Furthermore, as shown in Figure 1-(a), existing retrieval models often fail to identify and retrieve documents that effectively support decision-making.
>
> To address these issues, our research proposes a novel calibration method tailored to **RAG scenarios**, with the following two primary objectives:
> 1. Effectively identify and retrieve documents that support decision-making
> 2. Provide calibrated confidence scores that appropriately reflect uncertainty
>
> To achieve this, we developed a unique synthetic dataset specifically designed to evaluate decision calibration in RAG. Additionally, we simplified the training process by constructing a forecasting function using LLMs as encoders, requiring fine-tuning of only a single head. Furthermore, we leveraged the human evaluation results from Zhou et al. (2024) to introduce a surrogate model, enabling us to approximate human decision-making based on prompts without additional model training.
>
> This study aims to demonstrate that even complex problems such as decision calibration can be addressed in a simple, theoretically sound, and accessible manner. We believe that novelty does not solely rely on complex techniques. Despite its simplicity and ease of implementation, our method outperforms existing baselines and effectively addresses real-world challenges. Moreover, we emphasize that introducing a straightforward yet novel approach to tackle a previously unaddressed problem is, in itself, a significant contribution to the field.

---

> ### Author Response · Authors · 2024-11-20
>
> >**[w3] Key technical details, such as the cross-encoder baseline and the sensitivity of the LLM-rerank baseline, are insufficiently described, making the results less credible.**
>
> It is mentioned in Appendix A for more experimental details. For the sentence cross-encoder, we used the [stsb-TinyBERT-L-4](https://huggingface.co/cross-encoder/stsb-TinyBERT-L-4) linked in the paper and followed the same approach as the code snippet provided in [BEIR reranking](https://github.com/beir-cellar/beir/blob/main/examples/retrieval/evaluation/reranking/evaluate_bm25_ce_reranking.py). We apologize for not specifying these details clearly in the main text. We will provide a more detailed explanation in the Appendix.

---

> ### Author Response · Authors · 2024-11-20
> **Summary of Revisions**
>
> ---
>
> 1. Clarification of the use case and process of decision calibration **[W1]**
> 2. Clarification of section 2.1 **[W2]**
> 3. Key technical details **[W3]**
>
> ---
>
> **Your thoughtful questions could significantly improve the quality of our manuscript. Once you review and approve the planned updates, we will integrate them into the paper and clearly specify the sections where the corresponding changes have been made.**

---

> ### Author Response · Authors · 2024-11-23
>
> With the discussion deadline approaching, we have worked hard to provide clear and detailed answers to your questions. We are ready to make any necessary changes to the manuscript and would like to check if our responses have addressed your concerns. If you have any further questions or suggestions, we would greatly appreciate your feedback to help us improve our work. Thank you.
>
> ---
>
> **Given the time constraints, we have preemptively included the changes in the manuscript and provided a summary in the general response for your reference.**

---

> ### Comment · Reviewer_46aU · 2024-11-29
> **Thanks for the reply**
>
> Thanks the reply. After reviewing the reply and reviews from other reviewers, I decide to keep the score given the overall quality of the work.

---

> ### Author Response · Authors · 2024-11-30
>
> Thank you for your response. As mentioned in our general response, **we have addressed all the reviewers' comments and incorporated the suggested changes into the manuscript. Therefore, we find the statement regarding the "quality of the work" somewhat ambiguous.** If there are any specific limitations in our work that we could improve further, we kindly ask you to clearly highlight them. Otherwise, if your concerns have been adequately addressed, we would appreciate it if you reflect this in your scoring. Thank you for your consideration.

---

### Official Review · Reviewer_fXeq · 2024-11-01

**Soundness:** 3
**Presentation:** 2
**Contribution:** 2
**Rating:** 5
**Confidence:** 3

**Summary:**

This work focus the problem that the confidently incorrect information provided by LLMs may mislead human decision-making process. The target is to calibrate confidence of LLM responses with the correctness of decision-making based on these responses. The proposed CalibRAG framework, which contains synthetic supervision data generation and training, show notable improvement in confidence calibration and reranking tasks.

**Strengths:**

- It is a important research topic to calibrate LLM confidence in a retrieval-augmented generation (RAG) system. Confidence calibration of RAG is highly relevant for building trustworthy LLM systems, as in practical applications, LLMs are typically used with input knowledge.
- The proposed method demonstrates significant improvements in the uncertainty calibration task. The experimental setup includes diverse datasets, multiple baselines, and several standard evaluation metrics.

**Weaknesses:**

- The paper appears to equate "the confidence of LLM response" with "the relevance between the query and the document."
	- During training (Section 3.2), the forecasting function relies solely on information from the query and the document, excluding the actual LLM-generated guidance ($z$). Since the LLM-generated guidance may omit or alter important information from the document, using only the query and document information to substitute for the LLM-generated guidance is unreasonable. Therefore, the proposed method seems more akin to training a specialized rerank model rather than calibrating the confidence of the LLM response.
- The reranking experiments are not entirely fair due to the use of additional fine-tuning data compared to baseline methods. Moreover, the synthetic supervision data uses the correctness of user decisions (surrogate model in this paper) as the supervision signal, which results in a smaller gap and aligns more closely with the testing procedure compared to baseline methods.
- The results in Section 4.3 do not convincingly that the LLM can effectively simulate human decision-making. In the 0-20 and 40-60 bins, the agreement rate is only approximately 60%, indicating that a substantial portion of the data cannot be simulated by the LLM. Considering that in Table 2, the proposed method improves accuracy by about 2% compared to LLM-rerank on the BioASQ dataset, the approximately 30% disagreement between the LLM and human decision-making is significant.
- Figure 3 should be a vector image (import as pdf/svg) to ensure high quality.

**Questions:**

* Are there any fundamental differences or novel issues in confidence calibration for Retrieval-Augmented Generation (RAG) compared to calibration in generation models without retrieval augmentation?

---

> ### Author Response · Authors · 2024-11-20
>
> >**[W1] Using only the query and document information to substitute for the LLM-generated guidance is unreasonable.**
>
> Thank you for your insightful question. This is a very important question and was a key consideration in our design of CalibRAG. We recognized that predicting $ c  $ using only $ q $ and $ d $ without $ z $ is challenging. However, generating $ z $ for every $ q $-$ d $ pair and re-ranking based on it would impose a significant computational burden, making it impractical.
>
> To address this, we designed a linear head for prediction attached to the LLM that generates $ z $, allowing the LLM to use its own representations of $ q $ and $ d $ to predict $ c $. In other words, we are implicitly utilizing the knowledge of the LLM to judge whether the document $d$ is helpful for decision-making involving the query $q$. If this design were insufficient for predicting $ c $ with only $ q $ and $ d $, the model would not be able to learn or generalize effectively. However, our design has enabled the model to learn effectively.
>
> Additionally, to prevent extreme errors in LLM predictions, we included a prompt in data generation that instructed it to answer the query based on the document, ensuring consistency. Specific examples of this can be found in Appendix E, Table 3.
>
> However, we realize that by not emphasizing these design choices in the paper, it may have appeared that our approach assumes predicting $ c $ solely based on $ q $ and $ d $ as a given. We will add this clarification in the revised version to prevent any misunderstanding.
>
> >**[W2] Reranking experiments are not entirely fair due to the use of additional fine-tuning data compared to baseline methods.**
>
> Thank you for the critical question. First, all of our evaluations are conducted in a zero-shot setting. This means that the training and evaluation data differ significantly in both domain and composition, with the training data consisting primarily of synthetic data that we generated. Further details on this setup can be found in Appendix A.1. If we were to fine-tune the reranking baselines in the same way as our method, it would be very similar to our approach. Please note that the synthetic data generation process is also one of our contributions.  Moreover, many recent reranking studies also do not apply separate fine-tuning to their baselines [1,2,3].
>
> [1] Kim, Jaehyung, et al. "SuRe: Summarizing Retrievals using Answer Candidates for Open-domain QA of LLMs." The Twelfth International Conference on Learning Representations. (2024)
>
> [2] Cheng, Xin, et al. "Lift yourself up: Retrieval-augmented text generation with self-memory." Advances in Neural Information Processing Systems 36 (2024).
>
> [3] Askari, Arian, et al. "Expand, highlight, generate: RL-driven document generation for passage reranking." Proceedings of the 2023 Conference on Empirical Methods in Natural Language Processing. 2023.
>
> >**[W3] With only about 60% agreement in certain bins and a 2% accuracy improvement on BioASQ, how can we claim that the LLM effectively simulates human decision-making?**
>
> We agree that an LLM cannot perfectly mimic human behavior at 100%. However, having multiple human annotators label all evaluation data would incur substantial costs, and our annotator pool was limited in size and contained some outliers. Additionally, in cases where the LLM made decisions independently without our prompt imitating human decision-making, there was a significant gap between human and surrogate model decisions. We applied the best possible methods to minimize this gap, as demonstrated in Table R.6, where our approach shows effectiveness in reducing the gap.
>
> __Table R.6__  Comparison of Agreement Rates with Human Decisions Across Confidence Ranges
> | Range (%)  | Agreement Rate (With Prompt) | Agreement Rate (Without Prompt) |
> |------------|------------------------------|----------------------------------|
> | 0-20       | 70%                         | 30%                             |
> | 20-40      | 80%                         | 70%                             |
> | 40-60      | 60%                         | 40%                             |
> | 60-80      | 96.67%                      | 96.67%                          |
> | 80-100     | 100%                        | 100%                            |
> | **Average**| **81.33%**                  | **67.33%**                      |
>
>
> >**[W4] Figure 3 should be a vector image (import as pdf/svg) to ensure high quality.**
>
> We will update Figure 3 to a vector format (PDF/SVG) in the final version to ensure optimal quality.

---

> ### Author Response · Authors · 2024-11-20
>
> >**[Q1] Are there any fundamental differences or novel issues in confidence calibration for RAG compared to calibration in generation models without retrieval augmentation?**
>
> First, thank you for raising concerns and providing your questions. We have addressed the importance of our paper in the **general response section**, and we kindly ask you to refer to it for more details.
>
> In RAG, additional context that the LLM may not know is augmented into the input, which differs from the process where the LLM generates responses solely based on pre-existing knowledge or a given answer. This additional context serves as a hint, creating a different scenario compared to the traditional tasks performed by generation models.
>
> For instance, as shown in Table 1-(b), simple confidence calibration methods that do not account for RAG are insufficient to address these challenges. This is because, while RAG can improve performance, it can also lead to overly high confidence. However, existing research has not addressed decision calibration within the RAG framework.
>
> Moreover, if the LLM generates responses based solely on the Top-1 document retrieved by the retrieval model, it may fail to provide the optimal information required for decision-making. As illustrated in Figure 1-(a), other documents within the Top-10 may offer more valuable insights or information that contribute to more accurate decisions. This highlights the necessity of not only employing a retrieval-based approach but also integrating LLM-retrieval interactions and calibrating the confidence of the retrieved documents.
>
> Therefore, confidence calibration in RAG involves fundamentally different challenges and issues compared to calibration in simple generation models. It requires a comprehensive approach that not only enhances performance through RAG but also addresses the problem of over-confidence and compensates for incomplete or inaccurate information provided by the retrieval model.

---

> > ### Comment · Reviewer_fXeq · 2024-11-29
> >
> > Thanks for your response.
> > After reviewing your responses and considering the comments from other reviewers, I have decided to keep my original score.

---

> > > ### Author Response · Authors · 2024-11-30
> > >
> > > Thank you for your response. It seems that we have addressed all the questions raised by other reviewers, and as mentioned in our general response, we have already incorporated all the necessary changes into the manuscript. In fact, some reviewers have acknowledged that their concerns have been fully resolved. However, we are curious to know which aspects still remain a concern for you. If you could clarify your concerns in more detail, it would greatly help us further improve the manuscript. We truly appreciate your time and effort in providing feedback.

---

> ### Author Response · Authors · 2024-11-20
> **Summary of Revisions**
>
> ---
>
> 1. Discussion on substituting LLM-generated guidance **[W1]**
> 2. Comparison without a human prompt **[W3]**
> 3. Import Figure 3 as PDF/SVG **[W4]**
>
> ---
>
> **Your insightful questions have the potential to enhance the quality of our manuscript. Once you review and confirm the list of updates we plan to include, we will incorporate them into the paper and clearly indicate the sections where the corresponding changes have been made.**

---

> ### Author Response · Authors · 2024-11-23
>
> The discussion deadline is approaching, and we have made every effort to provide clear and comprehensive responses to your questions. We are prepared to make any necessary revisions to the manuscript and would like to confirm whether our answers have sufficiently addressed your concerns. Should you have any further questions or feedback, we kindly invite your input to help us enhance our work. Thank you.
>
> ---
>
> **We have outlined our first revised manuscript in the general response for your reference.**

---

### Official Review · Reviewer_wbiP · 2024-11-04

**Soundness:** 2
**Presentation:** 2
**Contribution:** 2
**Rating:** 3
**Confidence:** 4

**Summary:**

This paper proposes the Calibrated Retrieval-Augmented Generation (CalibRAG) framework. CalibRAG allows an LLM using RAG to not only select relevant information to support user decision-making but also provide confidence levels associated with that information by utilizing a forecasting function, ensuring well-calibrated decisions based on the retrieved documents.

**Strengths:**

1. Proposes a new dataset by creating labels that indicate whether decisions made using retrieved documents correctly answer the questions, essential for training the forecasting function.

2. The proposed method outperforms existing uncertainty calibration baselines across various tasks involving RAG context using the Llama-3.1 model in decision-making scenarios.

**Weaknesses:**

1. Many important baselines in RAG that have the same motivation with this paper are missing in experiments. This paper is actually aims to improve the robustness of RAG, which can select useful documents and prevent interference from noisy documents. So many important baselines such as RetRobust [1], Self-RAG [2], CRAG [3] should be discussed and compared but this paper overlooks them.

2. Technical contributions are limited. This paper uses a surrogate model that predicts the probability of whether the user’s decision based on the guidance provided by RAG will be correct. Many existing methods are similar to this such as Self-RAG [2] and CRAG [3]. I cannot learn any insightful points from this paper that is distinguishable enough from existing methods.

3. The writing should be improved. The Introduction section has so many contents about the existing methods that have been well-known in RAG area while there is too little contents about the methods and contributions for this paper.

4. The term "Decision Making" is too broad to describe only the evidence selection task in open-domain question answering. It is typically used in a much wider context.

[1] Making Retrieval-Augmented Language Models Robust to Irrelevant Context

[2] Self-RAG: Learning to Retrieve, Generate, and Critique through Self-Reflection

[3] Corrective Retrieval Augmented Generation

**Questions:**

Please refer to the details in the Weakness section:

1. Human decision-making is far more complex than described in this paper. How can we assume that if LLMs can handle document selection, they are also capable of decision-making?

2. How do other related baselines perform?

---

> ### Author Response · Authors · 2024-11-20
>
> >**[W1, W2, Q2] What are the baselines in RAG that share the same motivation as this paper, such as cRAG, selfRAG, and RetRobust?**
>
> We sincerely appreciate your valuable feedback. After carefully reviewing the suggested baselines, including Self-RAG and CRAG, we conducted additional experiments to analyze these methods more comprehensively. However, it is important to clearly explain the fundamental differences between our approach and these baselines.
>
> Our approach focuses on **predicting the accuracy of user decisions ($y$) based on the information ($z$) generated through RAG** and providing reliable calibration ($c$) during the RAG process. This fundamentally differs from Self-RAG, CRAG, and RetRobust, which focus on evaluating the relevance between the input query ($q$) and the document ($d$).
>
> Self-RAG introduces a utility score, which is defined *"as the perceived usefulness and informativeness of an answer to the query, regardless of whether the response is factually accurate. This aligns with the concept of plausibility discussed in Menick et al. (2022)."*  [1] CRAG evaluates the "correctness" and "incorrectness" of documents ($d$) for queries ($q$) based on their relevance. RetRobust learns whether $q$ can be inferred from $d$.
>
> These methods evaluate how well a document supports a query and classify results based on specialized tokens. However, such correctness or utility relies on ambiguous concepts and heuristics. In contrast, our method explicitly models **the accuracy of user decisions** based on generated responses and aims to achieve theoretically grounded calibration by maximizing **proper scoring rules** (e.g., likelihood) [2]. This distinction highlights that the two approaches address fundamentally different problems, making direct comparisons challenging. We apologize for not clarifying these differences earlier and will address this by summarizing the distinctions in the Appendix.
>
> Additionally, when attempting to interpret the utility scores of Self-RAG as a proxy for decision confidence, we identified the following limitations (refer to Table R.5 for details):
>
> 1. Independence of Scores
>    The utility score in Self-RAG is not inherently related to decision accuracy but focuses on "plausibility." As a result, it does not naturally align with metrics required to evaluate calibrated confidence, such as ECE, NLL, or Brier Score.
>
> 2. Practical Challenges
>    In experiments, Self-RAG tended to assign the highest utility score (e.g., utility:5, relevant token) to most top-ranked documents, making precise reranking of candidate documents practically difficult.
>
> 3. Performance Gap
>    Even after mapping utility scores to probabilities (e.g., mapping 1 to 0.2, etc.), significant performance gaps remained. The decision calibration performance of Self-RAG was consistently lower than ours across all datasets.
>
> Despite our efforts to align these baselines with our framework, the inherent differences in goals and methodologies limit the validity of direct comparisons. However, Self-RAG and CRAG have their own merits. For instance, Self-RAG is effective at determining when to retrieve documents, while CRAG can supplement missing information using web searches.
>
> In this context, we plan to include additional experiments and use cases in the Appendix to demonstrate the potential synergies when combining these baselines with CalibRAG.
>
> __Table R.5__: Comparison of CalibRAG and SelfRAG in Zero-Shot Decision Calibration
> | Method     | Dataset   | Acc      | AUROC   | ECE     | NLL     | BS      | NA      |
> |------------|-----------|----------|---------|---------|---------|---------|---------|
> | SelfRAG    | BioASQ    | 30.15 ± 0.07 | 50.00 ± 0.04 | 0.6932 ± 0.07 | 23.94 ± 0.08 | 0.6932 ± 0.09 | 1.72 ± 0.09  |
> |            | HotpotQA  | 33.92 ± 0.10 | 50.11 ± 0.01 | 0.6507 ± 0.00 | 22.43 ± 0.09 | 0.6505 ± 0.04 | 2.77 ± 0.08  |
> |            | WebQA     | 38.83 ± 0.02 | 50.00 ± 0.09 | 0.6104 ± 0.03 | 21.07 ± 0.04 | 0.6104 ± 0.11 | 3.00 ± 0.02  |
> |            | NQ        | 34.97 ± 0.09 | 50.09 ± 0.04 | 0.6471 ± 0.10 | 22.33 ± 0.07 | 0.6469 ± 0.05 | 8.00 ± 0.05  |
> | CalibRAG   | BioASQ    | 35.03 ± 0.14 | 71.21 ± 0.83 | 0.2500 ± 0.01 | 0.7899 ± 0.01 | 0.2900 ± 0.01 | 27.31 ± 0.97 |
> |            | HotpotQA  | 39.28 ± 0.79 | 65.47 ± 0.94 | 0.2414 ± 0.03 | 0.8276 ± 0.05 | 0.2876 ± 0.01 | 26.85 ± 2.12 |
> |            | WebQA     | 44.32 ± 0.42 | 69.58 ± 0.56 | 0.3064 ± 0.04 | 0.8919 ± 0.08 | 0.3251 ± 0.02 | 6.65 ± 0.73  |
> |            | NQ        | 48.29 ± 0.59 | 66.36 ± 0.68 | 0.2596 ± 0.02 | 0.8490 ± 0.01 | 0.2994 ± 0.02 | 8.39 ± 0.54  |
>
>
> [1] Asai, Akari, et al. "Self-RAG: Learning to Retrieve, Generate, and Critique through Self-Reflection." The Twelfth International Conference on Learning Representations.
>
> [2] Mielke, Sabrina J., et al. "Reducing conversational agents’ overconfidence through linguistic calibration." Transactions of the Association for Computational Linguistics 10 (2022).

---

> ### Author Response · Authors · 2024-11-20
>
> >**[W3] The term "Decision Making" is too broad to describe only the evidence selection task in open-domain question answering. It is typically used in a much wider context.**
>
> Thank you for your insightful question.  We also agree that in real-world scenarios, the term “Decision-making” can be bound to a broader range of contexts beyond open-domain question answering. However, it is challenging to accurately create real-world decision-making tasks in a synthetic setting. Previous studies have also restricted their focus to question-answering tasks, and we followed this approach [1,2,3]. We acknowledge that the point you raised is an important one, and we believe it is worth exploring further in future work. However, addressing this would go beyond the current scope of our study.
>
> [1] Band, Neil, et al. "Linguistic Calibration of Longform Generations." Proceedings of the Forty-first International Conference on Machine Learning. 2024.
>
> [2] Stengel-Eskin, Elias, Peter Hase, and Mohit Bansal. "LACIE: Listener-Aware Finetuning for Calibration in Large Language Models." Proceedings of the Thirty-eighth Annual Conference on Neural Information Processing Systems. 2024.
>
> [3] Mielke, Sabrina J., et al. "Reducing Conversational Agents’ Overconfidence through Linguistic Calibration." Transactions of the Association for Computational Linguistics, vol. 10, 2022, pp. 857–872.
>
>
> >**[Q1] Human decision-making is far more complex than described in this paper. How can we assume that if LLMs can handle document selection, they are also capable of decision-making?**
>
> Thank you for your insightful question. We agree that individual human decision-making is highly complex. Each person's decisions are shaped by numerous unpredictable factors, much like independent random variables with considerable variability. However, just as patterns can emerge when multiple random variables are combined, it is possible to approximate collective decision-making tendencies even if we cannot fully model each individual's decision-making process [1,2].
>
> Zhou et al. (2023) illustrated the general decision-making patterns within human groups. It included findings that reliance on LLM outputs increases sharply as the confidence expressed by the LLM becomes higher and plain text is regarded as a confident one. Building on this, CalibRAG constructs prompts that reflect these collective tendencies. This design allows the variance in LLM-generated responses to approximate the variability observed in human group decisions. Figure 3-(a) supports this, showing high agreement between decisions from our surrogate model and those of human annotators.
> To further validate that our prompt better captures human-like decisions compared to an LLM without it, we provide additional experimental results in Table R.6.
>
> __Table R.6__  Comparison of Agreement Rates with Human Decisions Across Confidence Ranges
> | Range (%)  | Agreement Rate (With Prompt) | Agreement Rate (Without Prompt) |
> |------------|------------------------------|----------------------------------|
> | 0-20       | 70%                         | 30%                             |
> | 20-40      | 80%                         | 70%                             |
> | 40-60      | 60%                         | 40%                             |
> | 60-80      | 96.67%                      | 96.67%                          |
> | 80-100     | 100%                        | 100%                            |
> | **Average**| **81.33%**                  | **67.33%**                      |
>
> [1] Tversky, Amos, and Daniel Kahneman. "Advances in prospect theory: Cumulative representation of uncertainty." Journal of Risk and Uncertainty 5 (1992): 297-323.
>
> [2] Acuna, Daniel, and Paul R. Schrater. "Structure learning in human sequential decision-making." Advances in neural information processing systems 21 (2008).

---

> ### Author Response · Authors · 2024-11-20
> **Summary of Revisions**
>
> ---
>
> 1. Experimental results comparing SelfRAG and CalibRAG **[W1]**
> 3. Clarification of terms related to decision-making **[W2]**
> 4. Discussion on whether LLMs can replace decision-making **[Q1]**
> 5. Experimental use cases combining CalibRAG with other robust retrieval methods (to be added if feasible) **[W1]**
>
> ---
>
> **Thank you for contributing to the improvement of our manuscript. Once you confirm these updates, we will integrate them into the paper and specify the sections where the corresponding changes have been implemented.**

---

> ### Author Response · Authors · 2024-11-23
>
> The discussion deadline is approaching. We have done our best to prepare thoughtful and thorough responses to your questions. We would like to confirm if our answers have sufficiently addressed your concerns. If you have any additional questions for discussion, we sincerely encourage your active participation to help us improve our research. Thank you.

---

> ### Author Response · Authors · 2024-11-24
>
> Due to limited time, we have proactively incorporated the changes into the manuscript and summarized them in the general response for your reference.

---

### Official Review · Reviewer_tAsW · 2024-11-09

**Soundness:** 2
**Presentation:** 2
**Contribution:** 2
**Rating:** 6
**Confidence:** 2

**Summary:**

In this paper, the authors build upon Band et al.'s (2024) work on linguistic calibration for long-form generation by designing an LLM-based Calibrated Retrieval-Augmented Generation framework. This framework ensures that decisions informed by the retrieved documents are well-calibrated. The authors validated the effectiveness of their approach in four QA datasets.

**Strengths:**

1. The model design in the paper appears well-constructed and reasonable.
2. Introducing a confidence score in LLM responses can effectively support human decision-making.

**Weaknesses:**

1. Section 2 lacks coherent organization, which limits its ability to effectively link the relevant background to the authors’ contributions. For example, the three issues identified at the end of Section 2.1 are not clearly addressed in the proposed approach. It would be helpful to clarify how this approach resolves each of these problems. Additionally, when formalizing the problem in Section 2.1, the authors should explicitly outline its relationship to previous work. Without this context, researchers in automated QA may struggle to understand the distinction between the original question and the open-ended query, as well as the necessity of introducing the open-ended query. Additionally, Section 2.2 should provide a clearer explanation of the calibration error, as it is challenging to intuitively grasp how performance differences impact decision-making outcomes when interpreting Figure 1. Moreover, while Figure 1a illustrates the effectiveness of calibRAG, Figure 1b does not, and despite the presentation of calibRAG’s results in Figure 1a, there is no corresponding description in the text.
2. The authors should expand the discussion on open-ended queries. Specifically, is it necessary for users to create these queries, or could they be generated by the LLM itself? What impact would LLM-generated queries have on the outcomes? Additionally, examining how variations in open-ended queries influence model performance would add valuable insights.
3. The authors incorporate confidence scores in long-form generation to enhance human decision-making. However, there is insufficient empirical analysis to demonstrate this benefit—specifically, a comparison of human judgment accuracy with and without confidence scores is missing. Automated evaluation alone is inadequate to validate this claim. Moreover, while the authors prompt the LLM to express uncertainty using either an integer scale from 0 to 10 or linguistic terms of certainty, the impact of these differing methods on model performance remains unclear. Further analysis is needed to clarify how these approaches influence outcomes.
4. The paper lacks comprehensive experimental validation. Evaluating the model on the complete BEIR benchmark would provide a more thorough assessment. Additionally, how does the model perform in non-zero-shot scenarios? For instance, is the hyperparameter setting—particularly the number of retrieved passages (K)—consistent across datasets, or does it vary? Stage 3 in Section 3.4 requires a more detailed description, specifically regarding how the query is reformulated in the experiments. Furthermore, how is the predefined threshold ϵ selected, and what effect does its value have on performance? The dataset verified by human annotators is too limited to robustly support the conclusions drawn from the experimental analysis. It would also be beneficial for the authors to showcase the performance of different models on human-annotated data.

**Questions:**

Please refer to the weaknesses.

---

> ### Author Response · Authors · 2024-11-20
>
> >**[W1] Clearly explain how the proposed approach addresses the three issues presented in Section 2.1.**
>
> **(Please review the general response, as it will help you better understand our paper.)**
>
> Thank you for your question regarding how we addressed the three key limitations of Band et al. (2024). As discussed in Section 2.1, the three key limitations are as follows: 1) the method requires supervised fine-tuning of three separate LLMs, 2) it additionally relies on PPO for fine-tuning the LLM used for response generation, 3) it cannot calibrate the probabilities associated with user decisions based on guidance provided by RAG.
>
> To address the first two limitations, which highlight the computational and methodological complexity of fine-tuning three LLMs and applying PPO, we developed a more efficient and streamlined approach. By introducing a lightweight linear layer to the LLM, we enable the inclusion of our forecasting function. This modification allows us to conduct supervised learning using a newly created dataset in a simple logistic regression, avoiding the need for multiple fine-tuning processes and policy optimization.
>
> For the third limitation, which pertains to the inability of Band et al. (2024) to handle the overconfidence of LLM-generated guidance in RAG-based scenarios, we proposed the CalibRAG method. This approach directly addresses the overconfidence issue by recalibrating the probability outputs of the LLM to better reflect the true uncertainty in the guidance. Our experimental results demonstrate that CalibRAG achieves well-calibrated results, even in complex RAG-based scenarios, and outperforms baseline methods. These improvements enhance the practical applicability of our method while overcoming the challenges in Band et al. (2024).
>
>
> >**[W2] The explanation of the figure in Section 2.2 is insufficient; while Figure 1a demonstrates the effectiveness of CalibRAG, Figure 1b does not. A clear explanation of calibration errors is needed.**
>
> Thank you for the constructive comment requesting a deeper explanation of Figure 1b. The purpose of this figure is to illustrate the observed effects of using RAG as guidance for task performance, specifically in the context of the NaturalQA dataset evaluated with Llama-3.1-8B. When we compare the performance of the standalone LLM model versus the same model with additional RAG guidance, we observe an improvement in accuracy when RAG guidance is included. However, the results also reveal that the inclusion of RAG guidance leads to a significant increase in the number of answers with high confidence levels (ranging between 0.8 and 1.0). This increase in overconfident predictions results in a notable degradation of calibration error.
>
> This experiment supports our argument introduced in the paper: while RAG guidance can enhance task accuracy, it also introduces a risk of over-reliance on the provided guidance, causing the model to produce overconfident predictions. This serves as a key motivation for our work, emphasizing the necessity of developing a model that remains well-calibrated even when leveraging RAG-based scenarios. Figure 1b, therefore, highlights both the potential benefits and drawbacks of RAG, providing clear evidence for why calibration in such settings is an essential and challenging problem.
>
>
>
> >**[W3] Is it necessary for users to create these queries, or could they be generated by the LLM itself? What impact would LLM-generated queries have on the outcomes?**
>
> Thank you for raising a question about the generation of open-ended queries $q(x) $ related to a question $x$ and how $q(x) $ affects performance. Ideally, a human decision-maker could craft a well-suited $q(x) $ for $x$ to receive a high-quality response. However, in our framework, CalibRAG leverages in-context learning with LLMs to generate $ q(x) $ automatically. This is achieved by providing a few high-quality $ (x, q(x)) $ examples and prompting the LLM to create $ q(x') $ for a new input $ x' $, which is then used to retrieve documents via RAG.
>
> The quality of these generated queries $ q(x) $ indeed influences performance. Specifically, in CalibRAG's inference pipeline, at stage 3, if the prediction probabilities derived from the documents retrieved in stage 2 do not exceed a certain threshold, CalibRAG dynamically generates a new $ q(x) $, incorporating the previously generated $q(x)$ into the prompt for refinement, to retrieve a fresh set of top-K documents. This mechanism enhances the relevance of retrieved documents to $ x $, improving the guidance provided to the model.
>
> Empirically, we observe that this iterative query refinement improves performance. The results marked with a dagger in Table 1 demonstrate this effect, showing that CalibRAG achieves higher accuracy while maintaining low calibration error when using enhanced $ q(x) $. These results highlight the importance of adaptive query generation in ensuring both accurate and well-calibrated decision-making.

---

> ### Author Response · Authors · 2024-11-20
>
> >**[W4] A comparison of human judgment accuracy with and without confidence scores is missing. Automated evaluation alone is inadequate to validate this claim.**
>
> Thank you for asking about the effect of confidence on human judgment accuracy. As you correctly pointed out, analyzing how LLM-generated confidence, when provided with the model's response, impacts human decision-making is critical.  If we can prompt a LLM to imitate human decision-making, the performance improvement of CalibRAG with the LLM decision-maker will likely result in a similar improvement with a human decision-maker. To validate how the LLM can approximate human decision making, as written in lines 432-438, we have employed 10 human annotators to answer 10 questions with confidence level by CalibRAG and found that 81 % of their answers agree with the predictions of the LLM decision making. This human evaluation again shows the benefit of CalibRAG.
>
> Actually, as mentioned in the introduction, Zhou et al. (2024) have already conducted a thorough analysis in this area through extensive human evaluations. Their findings indicate that human users often over-rely on LLM outputs during decision-making processes, and this tendency increases in proportion to the model's confidence. This over-reliance can lead to inaccurate decisions, especially when the confidence levels are high but the model's predictions are not well-calibrated.
>
>
>
> >**[W5] Difference impact of number and linguistic verbalized confidence.**
>
> Thank you for your question regarding the different effects of linguistic confidence, which our approach used as a baseline. Before explaining further, we would like to clarify that CalibRAG does not utilize linguistic or number confidence. Instead, CalibRAG provides confidence based on probability predictions generated by the forecasting function, whereas verbalized confidence was employed only in the baseline models.
>
> Typically, verbalized confidence is expressed as a continuous number in the range of [0, 100] [1,2]. However, LLMs often struggle to fully understand the precise meaning of such numerical values [3]. To address this limitation, we explored alternative approaches: representing confidence as linguistic expressions (e.g., “likely”) or prompting the model to select from discrete numerical values ranging from 0 to 10. These two approaches were named **Linguistic** and **Number**, respectively, in the baselines, and detailed prompt designs can be found in Appendix E.
>
> In response to the reviewer’s suggestion for a deeper analysis, we conducted additional experiments to evaluate the effectiveness of verbalized confidence representations. Using the MMLU dataset and the Llama-3-8B model, we assessed the performance of three types of confidence representations: **continuous number**, **discrete number**, and **linguistic**. As shown in Table R.1, discrete number and linguistic confidence representations outperformed continuous number baseline. Notably, linguistic confidence proved to be beneficial by addressing the limitations of the model’s understanding of numerical relationships and improving the calibration process. Building on this finding, we aimed to compare our method against a stronger baseline. Since this is one of our unique findings, we will include it in the Appendix to provide additional value to the readers.
>
> __Table R.1__ Results of Verbalized Confidence Fine-Tune Evaluation on the MMLU Dataset using Llama-3-8B.
>
> | Case                  | ACC   | ECE    |
> |-----------------------|-------|--------|
> | Continuous-Number        | 43.63 | 0.3190  |
> | Discrete-Number    | 44.96 | 0.1605 |
> | Linguistic   | 45.03 | 0.1585 |
>
> [1] Tian, Katherine, et al. "Just Ask for Calibration: Strategies for Eliciting Calibrated Confidence Scores from Language Models Fine-Tuned with Human Feedback." The 2023 Conference on Empirical Methods in Natural Language Processing.
>
> [2] Xiong, Miao, et al. "Can LLMs Express Their Uncertainty? An Empirical Evaluation of Confidence Elicitation in LLMs." The Twelfth International Conference on Learning Representations.
>
> [3] Gambardella, A., Iwasawa, Y., & Matsuo, Y. (2024). Language models do hard arithmetic tasks easily and hardly do easy arithmetic tasks. Proceedings of the 62nd Annual Meeting of the Association for Computational Linguistics (Volume 2: Short Papers), 85-91. Association for Computational Linguistics. https://doi.org/10.18653/v1/2024.acl-short.8.

---

> ### Author Response · Authors · 2024-11-20
>
> >**[W6] Evaluating the model on the complete BEIR benchmark would provide a more thorough assessment.**
>
> Thank you for suggesting additional evaluations using the BEIR benchmark. We agree that such evaluations can further validate the effectiveness of CalibRAG. However, we would like to emphasize that CalibRAG is not just a reranking method trained to optimize model performance. Instead, it is designed for well-calibrated decision-making, focusing on predicting the probability $c$ of a correct decision when a user relies on the generated model guidance $z$ to make a decision about a given problem $x$.
>
> That said, based on your suggestion, we conducted experiments using two datasets from the BEIR benchmark: SciFact and TREC-COVID. We compared CalibRAG with the Cross-Encoder baseline, following the standard retrieval pipeline: retrieving documents with BM25 and reranking the top-100 results.
>
> The results, presented in Table R.2, clearly demonstrate that CalibRAG outperforms the Cross-Encoder. These findings validate that CalibRAG not only enables well-calibrated decision-making but also improves retrieval performance, thereby reinforcing its utility in relevant scenarios. Although we were only able to conduct experiments on two datasets due to time constraints, we promise to include evaluations on all datasets in the final manuscript.
>
> __Table R.2__ Evaluation result on TREC-COVID, SciFact datasets, a subset of BEIR benchmark. Evaluation metric is Normalized Discounted Cumulative Gain (NDCG@K).
>
> | Model            | Dataset        | NDCG@5     | NDCG@10    |
> |------------------|---------------|--------|--------|
> | Cross Encoder | TREC-COVID    | 0.7655 | 0.7576 |
> |                  | SciFact       | 0.6668 | 0.6914 |
> | CalibRAG         | TREC-COVID    | 0.7863 | 0.7660 |
> |                  | SciFact       |   0.6872     |   0.7114     |
>
>
>
>
> >**[W7] How does the model perform in non-zero-shot scenarios? For instance, is the hyperparameter setting—particularly the number of retrieved passages (K)—consistent across datasets, or does it vary?**
>
> Thank you for asking about whether the optimal number of retrieved passages varies across datasets. Through our experiments with the forecasting function and evaluations on various datasets, we observed that the optimal value of top $K$ can indeed vary depending on the dataset.
>
> For instance, in the case of the WebQA dataset, we found that retrieving only the top 20 documents and selecting the best document for inference resulted in lower performance when leveraging stage 3 of CalibRAG. In this stage, we not only utilized the original top 20 documents but also retrieved an additional 20 documents using a new query, which led to a significant performance improvement. On the other hand, for the NQ dataset, applying stage 3 to retrieve more documents did not lead to a noticeable improvement over selecting the best document from the original top 20. This suggests that for WebQA, using a larger $K$ is beneficial, while for NQ, a top 20 selection is sufficient and optimal.
>
> These observations highlight the importance of dataset-specific tuning for the number of retrieved passages to maximize performance. Since our evaluation setting is zero-shot, we determined the $K$ using the validation set from the synthetic dataset for fine-tuning. As shown in Appendix A.1, our synthetic d data covers a diverse range of domains. If a validation set exists, it would also be an alternative for determining K.

---

> ### Author Response · Authors · 2024-11-20
>
> >**[W8] How the query is reformulated in the experiments.**
>
> Thank you for asking about how queries are reformulated at stage 3 in CalibRAG.
>
> When generating the initial query, we simulate a situation where a human decision-maker crafts a simple query $q(x)$ related to the input $x$. This reflects a scenario where a decision-maker, facing a problem (e.g., "Is a tomato a fruit or a vegetable?"), might create a straightforward question such as "Classification of tomatoes" to query a language model. Using this setup, we employed an LLM generator to create queries that are simple yet relevant to the decision problem, and then retrieved documents based on these queries. Then if the retrieved documents were not sufficiently informative, the query was reformulated in stage 3. This reformulation emphasized key terms to refine the query and improve the quality of retrieved documents. The specific prompt used for this process is detailed in Appendix Table 8.
>
> To help readers better understand the transformation between the initial query and its reformulated version, we include a realistic example in Table R.3. This example will clearly illustrate how the query evolves through this refinement process, providing practical insight into the reformulation mechanism.
>
> __Table R.3__ Examples of Regenerated Queries
>
> | Case | Original Query                                                                                      | Regenerated Query                                                                                             | Explanation of Reformulation                                                                                                      |
> |----------|---------------------------------------------------------------------------------------------------------|-------------------------------------------------------------------------------------------------------------------|---------------------------------------------------------------------------------------------------------------------------------------|
> | 1        | Write a paragraph about the effect of TRH on myocardial contractility.                                  | Write a paragraph about the effect of Thyrotropin-Releasing Hormone (TRH) on myocardial contractility.            | Expanded "TRH" for clarity and formality.                                                                                             |
> | 2        | Write a paragraph the clinical trials for off-label drugs in neonates as cited in the literature.        | Write a paragraph about clinical trials for off-label drug use in neonates as reported in the medical literature. | Improved grammar and replaced "cited in the literature" with "reported in the medical literature" for precision.                      |
> | 3        | Write a paragraph about the current representatives from Colorado.                                      | Write a paragraph about the current representatives from the state of “Colorado” in the United States.            | Specified "state of Colorado in the United States" for clarity.                                                                       |
> | 4        | Write a paragraph about the current minister of local government in Zimbabwe and their role within the government. | Write a paragraph about the current Minister of Local Government and Public Works in Zimbabwe and their role within the government. | Updated the title to reflect its full official name.                                                                                  |

---

> ### Author Response · Authors · 2024-11-20
>
> >**[W9] How is the predefined threshold $\epsilon$ selected, and what effect does its value have on performance?**
>
> Thank you for pointing out the missing details. In our experiments, we set $\epsilon = 0.5$ as a balanced choice to manage the trade-off between accuracy and calibration error. As shown in Table R.4, increasing $\epsilon$ retrieves a larger number of new queries, incorporating more relevant information and thereby improving accuracy. However, this can also potentially increase the calibration error.  Specifically, while better retrieval led to more accurate predictions, the confidence scores for these predictions tended to increase only marginally. This discrepancy between improved accuracy and relatively low confidence resulted in underconfident samples, which appeared to slightly increase the calibration error.
>
> To evaluate the impact of different $\epsilon$ values on model performance, we conducted experiments on the BioASQ dataset. Based on these observations, we selected $\epsilon = 0.5$ as a reasonable compromise to balance accuracy improvements with calibration reliability.
>
> __Table R.4__  Effect of Threshold Selection on Performance.
> | Methods | AUROC        | ACC           | ECE            | BS             | NLL            | %NA           |
> |---------|--------------|---------------|----------------|----------------|----------------|---------------|
> | $\epsilon$=0.0 | 71.21 ± 0.83 | 35.03 ± 0.14  | 0.2500 ± 0.01  | 0.2900 ± 0.01  | 0.7899 ± 0.01  | 27.31 ± 0.97  |
> | $\epsilon$=0.4 | 76.15 ± 1.50 | 35.05 ± 0.25  | 0.2608 ± 0.00  | 0.2830 ± 0.00  | 0.7703 ± 0.03  | 26.57 ± 0.80  |
> | $\epsilon$=0.5 | 76.50 ± 4.98 | 35.98 ± 0.38  | 0.2667 ± 0.00  | 0.2779 ± 0.01  | 0.7560 ± 0.04  | 25.16 ± 0.42  |
> | $\epsilon$=0.6 | 77.20 ± 4.10 | 36.50 ± 0.45  | 0.2707 ± 0.00  | 0.2800 ± 0.01  | 0.7620 ± 0.03  | 24.98 ± 0.50  |
>
>
>
> >**[W10] Showcases the performance of different models on human-annotated data.**
>
> Thank you for suggesting a more thorough human evaluation. While we recognize the importance of high-quality human annotations, collecting experimental results from a large pool of human annotators incurs substantial costs. Instead, we focused on maximizing the agreement between human annotators and the model using carefully designed human prompts, leveraging this agreement for our overall evaluation. Specifically, by employing our human prompts, we improved the human vs. LLM agreement from 67.33% to 81.33% with CalibRAG. This improvement demonstrates that our model can effectively serve as a surrogate for human annotators to some extent. Additionally, using an LLM in this way offers greater reproducibility compared to human evaluation, which could benefit future experiments. We will include this result in the discussion section of the Appendix.

---

> ### Author Response · Authors · 2024-11-20
> **Summary of Revision**
>
> ---
>
> 1. Clarification in Sections 2.1 and 2.2  **[W1, W2]**
> 2. Additional analysis on verbalized baselines **[W5]**
> 3. Evaluation performed on the BEIR benchmark **[W6]**
> 4. Examples of query reformulation **[W8]**
> 5. Analysis of the effect of epsilon **[W9]**
> 6. Regarding the use of the LLM as a surrogate for humans **[W10]**
>
> ---
>
> **If you confirm these results, we will incorporate them into the manuscript and specify the sections where the corresponding updates have been made.**

---

> ### Author Response · Authors · 2024-11-23
>
> The discussion deadline is approaching. We would like to ask if we have addressed your concerns. If they have been resolved, please let us know so we can incorporate your decision into the manuscript. If not, we would be happy to continue the discussion with you. Thank you.

---

> ### Author Response · Authors · 2024-11-24
>
> We have summarized our first revised manuscript in the general response for your reference.

---

> > ### Comment · Reviewer_tAsW · 2024-11-26
> >
> > Thank you for your detailed response, which has addressed many of my concerns. However, considering the overall quality of the submission, I have decided to maintain my original score.

---

> ### Author Response · Authors · 2024-11-26
>
> Thank you for your valuable feedback. We have carefully reviewed all the points raised in the official review and incorporated the suggested improvements into the revised manuscript. As you also mentioned that many of your concerns have been addressed, we find it difficult to fully understand the specific reference to "quality of submission". Please let us know if there are any additional concerns or areas for improvement, and we will address them promptly. We believe this discussion process is crucial for advancing academic research, and we sincerely appreciate your constructive input.

---

> > ### Comment · Reviewer_tAsW · 2024-11-27
> >
> > I appreciate your efforts in revising the paper. I am willing to increase the score to 6.

---

> > > ### Author Response · Authors · 2024-11-30
> > >
> > > We sincerely thank you for raising your score and deeply appreciate your involvement in the process of improving our paper.

---

### Author Response · Authors · 2024-11-20
**General Response (1)**

We thank all reviewers for their valuable and constructive comments. **We are eager to engage in the review process and address the suggested areas for improvement to further strengthen the quality of this work.**

---

Before reviewing your respective rebuttals, we kindly ask all reviewers to read the following:

The ultimate purpose of calibrating LLMs is to **minimize the risks associated with user decisions** that rely on LLM-generated outputs. In highly sensitive domains such as medicine, confidently providing incorrect information can have severe results. Thus, the practical goal of calibration is to **ensure the model does not express high confidence in uncertain information**, enabling users to reconsider or postpone their decisions. This aligns with the concept of "decision calibration" as defined by Band et al.

However, recent open-source LLMs that incorporate retrieval-based methods have shown increased calibration error. As illustrated in Table 1-(b), existing calibration methods not designed for retrieval contexts lead to higher calibration error, even when retrieval contexts improve overall performance. Moreover, as shown in Table 1-(a), current RAG approaches fail to retrieve documents that help decision-making.

To address these challenges, we propose a novel calibration method tailored to RAG scenarios, with two primary objectives:

1. **Identify and retrieve documents that support decision-making**

2. **Provide calibrated confidence scores that reflect uncertainty appropriately**

To achieve these goals, we designed a system where the LLM as an encoder, paired with a classifier head explicitly models **the accuracy of user decisions** based on the LLM-generated responses. This classifier is trained to maximize **proper scoring rules** (e.g., likelihood), ensuring theoretically sound calibration. Unlike Band et al., who neither addressed RAG scenarios nor tuned classifiers for likelihood-based probability calibration, our approach introduces a fundamentally distinct methodology.

In addition, existing RAG variants (e.g., selfRAG, cRAG) primarily focus on **efficient operations** or providing **rich, contextually relevant information** for question answering, rather than directly addressing decision calibration. These approaches often define correctness based on utility scores (which may be unrelated to factual accuracy) or query-document relevance, relying on heuristic-driven and conceptually ambiguous metrics. Consequently, they fail to satisfy the theoretical requirements of proper scoring rules and differ fundamentally from our definition of calibrated confidence.

This study highlights the necessity of **calibration designed specifically for RAG** in LLM research and introduces a **simple yet effective solution**. Our method is compatible with any LLM encoder and utilizes synthetic training data explicitly crafted for this purpose, which we have made publicly available along with our code. As the **first approach to provide practical and theoretically grounded decision calibration in RAG scenarios**, we aim to establish a new standard for LLM research and its applications.

---
**We sincerely appreciate the constructive feedback once again, as it significantly enhances the robustness of our paper. We have summarized specific revision points at the end of each response to the reviewers' questions. If you find these points agreeable, we will incorporate them into the manuscript and notify you accordingly.**

---

### Author Response · Authors · 2024-11-24
**The Release of the First Revised Manuscript.**

https://openreview.net/pdf?id=nNQmZGjEVe

---

Here is how the feedback has been addressed in the manuscript. Due to the limited time, we incorporated the requested changes in advance:

* Updates to address the issues in Section 2 are highlighted in red within Section 3. Specifically:
    * (tAsW W1, W2)
    * (fXeq W1)
    * (46aU W2)
* The following changes have been applied to the tables in Appendix B:
    * Table 10: (tAsW W9)
    * Table 11: (tAsW W6)
    * Table 12: (tAsW W5)
    * Table 13: (tAsW W10, wbiP W2, fXeq W2)
    * Table 14: (wbiP W1)
* Additionally, adjustments were made to Appendix C:
    * Table 15: (tAsW W8)
* Replaced Figure 3 with a PDF image (fXeq W3).
---

We hope these updates meet the reviewers' expectations and look forward to receiving further feedback.

---

### Meta-Review · Area_Chair_vjC5 · 2024-12-24

**Metareview:**

This paper introduces a new retrieval method, named Calibrated Retrieval-Augmented Generation (CalibRAG), to ensure the decisions informed by the retrieved documents are well-calibrated. The authors have performed empirical experiments to demonstrate that CalibRAG improves both the calibration performance and accuracy.

Overall, this paper introduces some interesting ideas. However, the technical contribution of the proposed method seems limited. Some experimental settings are not fair enough (e.g., the rerankig experiments). The experimental analysis is not suffcient enough. Some important baselines are missing.

**Additional Comments On Reviewer Discussion:**

In the rebuttal, the authors provide more discussions about the figure in Section 2.2, the necessary of user queries, impact of the LLM generated queries, the impact of number and linguistic verbalized confidence, the reformulation of queries in the experiments, and the impact of the threshold \epsilon. Moreover, they also provide more experimental results of SelfRAG and CalibRAG, and the experimental results on BEIR benchmark dataset. Some concerns of the reviewers should have been addressed. However, their concerns regarding with the novelty of the proposed method still remain.

---

### Decision · Program_Chairs · 2025-01-22

Reject